# Glycosylphosphatidylinositol-anchored proteins as chaperones and co-receptors for FERONIA receptor kinase signaling in Arabidopsis

Chao Li[1], Fang-Ling Yeh[1†], Alice Y Cheung[1,2,3]*, Qiaohong Duan[1], Daniel Kita[1,2‡], Ming-Che Liu[1,4], Jacob Maman[1], Emily J Luu[1], Brendan W Wu[1§], Laura Gates[1¶], Methun Jalal[1], Amy Kwong[1], Hunter Carpenter[1], Hen-Ming Wu[1,2]*

[1]Department of Biochemistry and Molecular Biology, University of Massachusetts, Amherst, United States; [2]Molecular and Cell Biology Program, University of Massachusetts, Amherst, United States; [3]Plant Biology Graduate Program, University of Massachusetts, Amherst, United States; [4]Graduate Institute of Biotechnology, National Chung Hsing University, Tai Chung, Taiwan

*For correspondence: acheung@ biochem.umass.edu (AYC); hmwu@biochem.umass.edu (HMW)

Present address: [†]Clinical Research Center, Chung Shan Medical University Hospital, Taichung, Taiwan; [‡]Department of Vascular Biology, University of Connecticut Health Center, Farmington, United States; [§]Department of Immunology, Harvard Medical School, Boston, United States; [¶]Department of Biology, University of California at San Diego, La Jolla, United States

Competing interests: The authors declare that no competing interests exist.

**Abstract** The Arabidopsis receptor kinase FERONIA (FER) is a multifunctional regulator for plant growth and reproduction. Here we report that the female gametophyte-expressed glycosylphosphatidylinositol-anchored protein (GPI-AP) LORELEI and the seedling-expressed LRE-like GPI-AP1 (LLG1) bind to the extracellular juxtamembrane region of FER and show that this interaction is pivotal for FER function. LLG1 interacts with FER in the endoplasmic reticulum and on the cell surface, and loss of LLG1 function induces cytoplasmic retention of FER, consistent with transport of FER from the endoplasmic reticulum to the plasma membrane in a complex with LLG1. We further demonstrate that LLG1 is a component of the FER-regulated RHO GTPase signaling complex and that *fer* and *llg1* mutants display indistinguishable growth, developmental and signaling phenotypes, analogous to how *lre* and *fer* share similar reproductive defects. Together our results support LLG1/LRE acting as a chaperone and co-receptor for FER and elucidate a mechanism by which GPI-APs enable the signaling capacity of a cell surface receptor.

## Introduction

The Arabidopsis FERONIA (FER) receptor kinase critically controls growth and development, is indispensable for reproduction, and participates in defense-related responses (*Wolf and Hofte, 2014*). FER was initially identified as an essential regulator for female fertility (*Huck et al., 2003*; *Rotman et al., 2003*; *Escobar-Restrepo et al., 2007*; *Kessler and Grossniklaus, 2011*; *Duan et al., 2014*); its expression in the female gametophyte is responsible for inducing rupture of an invading pollen tube to release sperm for fertilization. It is also required to prevent supernumerary pollen tube entrance to individual ovules, precluding polyspermy and maximizing seed yield. Thus *fer* mutant plants are severely female-deficient, producing few seeds. FER is, however, broadly expressed and absent only in pollen (*Zimmermann et al., 2004*; *Duan et al., 2010*); its functions intersect several major plant hormone signaling pathways, including auxin (*Duan et al., 2010*), abscisic acid (ABA) (*Yu et al., 2012*), brassinosteroid, and ethylene (*Guo et al., 2009*; *Deslauriers and Larsen, 2010*). FER has also been shown to interact with the peptide hormone rapid alkalinization factor 1 (RALF1) (*Haruta et al., 2014*). Therefore *fer* knock-out mutants are pleiotropic, with vegetative phenotypes

**eLife digest** Plants respond to changes in their environment by altering how they grow and when they reproduce. A protein called FERONIA is found in most types of cells and regulates many of the processes that drive these responses, such as cell growth and communication between male and female cells. FERONIA sits in the membrane that surrounds the cell, where it can detect molecules in the cell wall and from outside the cell, and send signals to locations within the cell. However, it is not clear how FERONIA is able to specifically regulate different processes to produce the right response in a particular cell at a particular time.

A family of proteins called glycosylphosphatidylinositol-anchored proteins (GPI-APs for short) play important roles in plants, animals, and other eukaryotic organisms. Li et al. studied FERONIA and two closely related GPI-APs called LLG1—which is produced in seedlings, and LORELEI, which is only found in female sex cells. The experiments show that plants missing either LLG1 or FERONIA had similar defects in growth and in how they respond to plant hormones. Plants missing LORELEI had similar defects in their ability to reproduce as the plants missing FERONIA. This suggests that FERONIA works with either LLG1 or LORELEI to regulate similar processes in different situations.

Li et al. found that FERONIA binds to LLG1 in a compartment within the cell called the endoplasmic reticulum—where proteins are assembled—before both proteins are moved together to the cell membrane. In the absence of LLG1, FERONIA fails to reach the cell membrane, and a large amount of FERONIA remains trapped in the endoplasmic reticulum. Therefore, LLG1 acts as a 'chaperone' that delivers FERONIA to the membrane where it is required to regulate plant growth. Li et al. found that LORELEI also interacts with FERONIA. Both LLG1 and LORELEI bind to the same region of FERONIA, which is on the outer surface of the cell membrane.

These findings show that FERONIA is able to perform different roles in cells by teaming up with different members of the GPI-AP family of proteins. The next challenges will be to find out if, and how, LLG1 and LORELEI affect the ability of FERONIA to respond to signals from the cell wall and outside the cell.

attributable to defects in growth processes regulated by these hormones. FER also mediates susceptibility to the fungal pathogen powdery mildew (*Kessler et al., 2010*) and has been implicated in mechano-sensing (*Shih et al., 2014*). Thus it is likely that FER mediates distinct signals under different cellular and developmental conditions and environmental challenges.

How FER achieves its multiple functionality remains unclear. We showed earlier that FER interacts with ROPGEFs (*Duan et al., 2010*), guanine nucleotide exchange factors that stimulate GDP/GTP exchange in RAC/ROPs, the RHO GTPases of plants, activating them (*Berken et al., 2005*). We demonstrated that FER acts as a cell surface regulator for RAC/ROP-mediated NADPH oxidase-dependent reactive oxygen species (ROS) production to support polarized root hair growth in seedlings (*Duan et al., 2010*), and induce $Ca^{2+}$-dependent pollen tube rupture and sperm release in the female gametophyte (*Duan et al., 2014*). RAC/ROPs are known to mediate multiple signaling pathways that underlie normal plant growth and development, as well as stress-related responses (*Nibau et al., 2006*; *Wu et al., 2011*). ROS are ubiquitous and regulate a broad spectrum of cellular processes as diverse as cell growth and cell death (*Carol and Dolan, 2006*; *Jaspers and Kangasjarvi, 2010*; *Swanson and Gilroy, 2010*). Utilizing RAC/ROPs and ROS as signal mediators potentially provides almost limitless permutations of how FER-mediated signals might be propagated.

The diverse functionality of FER could also be provided by its potential ability to interact with multiple ligands. Its extracellular domain shows homology with malectin, a disaccharide-binding protein located in the endoplasmic reticulum (ER) of animal cells (*Schallus et al., 2008*, *2010*). That FER might interact with carbohydrate moieties suggests the potential of mediating cell wall perturbations elicited by a battery of endogenous and environmental conditions (*Hematy and Hofte, 2008*; *Boisson-Dernier et al., 2011*; *Cheung and Wu, 2011*; *Lindner et al., 2012*) such as hormonal changes impacting cell growth and pathogen attacks eliciting cell wall restructuring. RALF1 is one of ~40 related secreted peptides in Arabidopsis that collectively are ubiquitously present, albeit individually they are all expressed at low levels and their functional roles in plant growth and development remain largely unexplored (*Morato do Canto et al., 2014*; *Srivastava et al., 2009*).

If, similar to RALF1 (*Haruta et al., 2014*), more of these peptide hormones interact with FER, using individual RALFs as signals might be another strategy to achieve its multi-functional roles.

Glycosylphosphatidylinositol-anchored proteins (GPI-APs) are cell surface-located proteins known to play important roles in regulating a broad range of biological processes including growth, morphogenesis, reproduction, and disease pathogenesis in eukaryotes (*Lingwood and Simons, 2010*; *Fujita and Kinoshita, 2012*; *Yu et al., 2013*). They localize to sphingolipid- and cholesterol-enriched domains in the cell membrane where they are believed to play key roles in regulating cell surface signaling dynamics, although much remains to be learned about their precise functional mechanisms. In plants, GPI-APs play indispensable roles throughout development, required for cell wall biosynthesis, embryo viability, organogenesis, reproductive development, and male–female interactive processes crucial for fertilization (*Cheung et al., 2014*). LORELEI (LRE) and LRE-like GPI-APs 1, 2, 3 (LLG1, 2, 3) are closely related but differentially expressed (*Capron et al., 2008*; *Tsukamoto et al., 2010*). LRE is expressed exclusively in the ovule and loss of LRE function suppresses female fertility. *lre* mutants display reproductive phenotypes almost identical to those in *fer* mutants: a majority of *lre* and *fer* female gametophytes fail to induce rupture of the invading pollen tubes and their ovules are penetrated by multiple pollen tubes, yet fail to be fertilized because of the lack of sperm release. Here we show that LRE and LLG1 interact physically with FER and that they are crucial for its cell surface signaling capacity. Our results show partnering with related but differentially expressed proteins as a strategy for FER to execute its diverse biological roles; they also elucidate a novel mechanism for how GPI-APs might control cell surface signaling.

## Results

### *llg1* and *fer* mutants have indistinguishable growth and developmental phenotypes

Gene expression and mutant analyses showed that LLG1 is important for vegetative growth and development. LLG1 is the most prominent LRE family protein expressed in vegetative tissues (*Zimmermann et al., 2004*), where FER expression is also prevalent (*Duan et al., 2010*). The LLG1 promoter::GUS (*pLLG1::GUS*) expression pattern (*Figure 1A*; *Figure 1—figure supplement 1A*) overlapped considerably with that of *pFER::GUS* (*Duan et al., 2010*) in Arabidopsis seedlings (see *Supplementary file 1* for a list of constructs used). Two T-DNA-induced knock-out mutants, *llg1-1* and *llg1-2*, were indistinguishable from each other and from *fer-4*, a previously characterized knock-out *fer* mutant (*Duan et al., 2010*, *2014*) throughout vegetative development (*Figure 1B–E*; *Figure 1—figure supplement 1B,C*). Results described from here on are largely based on *llg1-2*, the mutant with which this work was initiated; observations made with *llg1-1* provided confirmation.

Similar to *fer-4*, *llg1* plants showed retarded growth starting from 4–5 days after germination and seedlings looked visibly stressed, accumulating higher levels of anthocyanin and appeared more purplish than wild type seedlings (*Figure 1C*, upper). Under dark-grown conditions, *llg1* were also de-etiolated and showed reduced apical hook bending relative to wild type, similar to *fer-4* (*Deslauriers and Larsen, 2010*) (*Figure 1E*; *Figure 1—figure supplement 1C*). Both *llg1* and *fer-4* remained smaller than wild type throughout growth and at maturity (*Figure 1C* lower; *Figure 1D*). Contrary to pronounced reproductive defects shared by *fer-4* and *lre* mutants, *llg1* plants had no reproductive phenotype and produced normal amounts of seeds, consistent with negligible LLG1 promoter activity in pollen and ovules (*Figure 1—figure supplement 1A*).

*llg1* mutants also developed root hair and trichome defects, similar to *fer* mutants. A large majority of *llg1* root hairs collapsed upon emergence and those that emerged remained significantly shorter than wild type root hairs (*Figure 2A,B*). Trichomes on *llg1* leaf epidermis were mostly defective, with a significant number of them having curly and more than three branches relative to those on wild type leaves (*Figure 2C*). Expression from a genomic LLG1 fragment and from a LLG1 promoter-expressed HA-tagged LLG1 (*pLLG1::HA-LLG1*) in *llg1-2* fully complemented its phenotypes (*Figure 2B,C*; *Figure 2—figure supplement 1*), confirming that loss of LLG1 function underlies the *fer*-like defects in *llg1* mutants. Furthermore, *fer-4 llg1-2* double mutant seedlings were indistinguishable from their single mutant parents (*Figure 2—figure supplement 2*). Together these results are consistent with FER and LLG1 functioning in the same pathways and that both proteins are required for these pathways, just as FER and LRE are both required to mediate reproductive success by controlling similar events in pollen tube–ovule interaction (*Capron et al., 2008*; *Tsukamoto et al., 2010*).

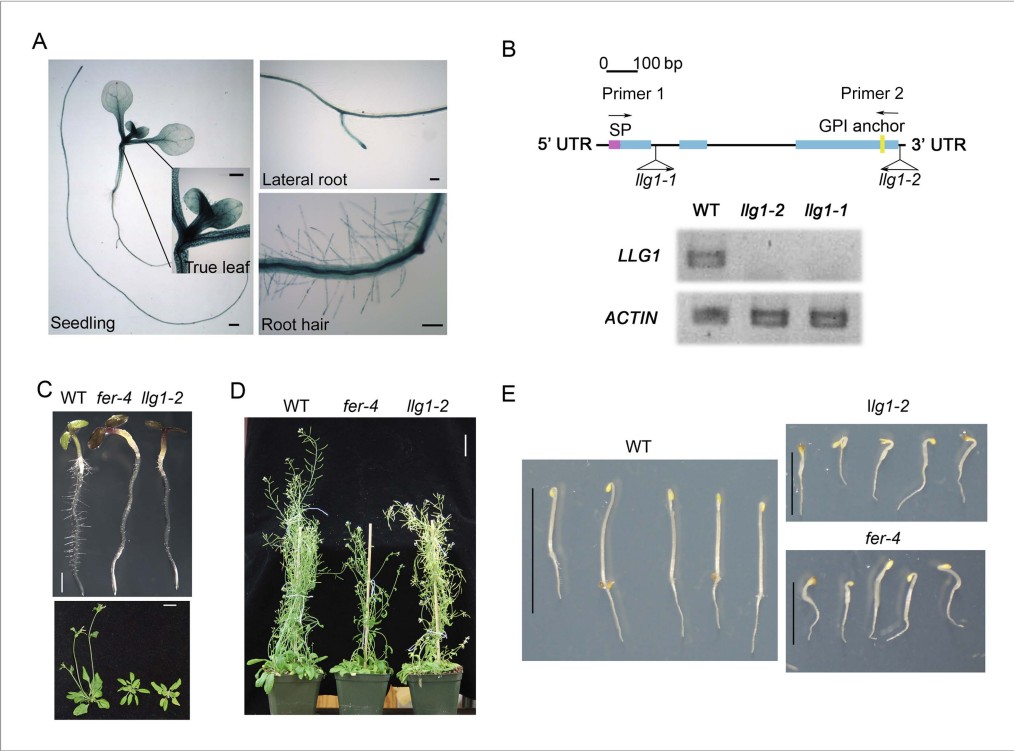

**Figure 1**. *llg1* and *fer* mutants have similar growth and developmental phenotypes. (**A**) *pLLG1::GUS* expression pattern in 10-day-old seedlings. (**B**) T-DNA insertion map for *llg1-1* and *llg1-2* (upper). RT-PCR analysis (using primers 1 and 2) indicates both mutants are nulls (lower). (**C**–**E**) Growth comparison between wild type (WT), *llg1* and *fer* plants. Four-day-old light-grown (**C** upper), 25-day-old (**C** lower), and flowering plants (**D**), and 3-day-old dark-grown seedlings (**E**). Scale bars: 1 mm (**A**); 2 mm (**C** upper); 3 cm (**C** lower, **D**); 1 cm (WT), 0.5 cm (*llg1-2*, *fer-4*) (**E**). See additional data in *Figure 1—figure supplement 1*.

The following figure supplement is available for figure 1:

**Figure supplement 1**. Additional characterization of *llg1* mutants.

## *llg1* and *fer* mutants have indistinguishable hormone- and RAC/ROP-regulated phenotypes

Given the role of FER in controlling RAC/ROP-regulated ROS production in seedling roots and that FER and LRE regulate ROS levels in the female gametophyte to mediate sperm release from the pollen tube (*Duan et al., 2010*), we examined whether loss of LLG1 conferred similar defects as those in *fer-4* seedlings. *llg1* root ROS levels were significantly reduced and did not respond to auxin-stimulated ROS accumulation as wild type did (*Figure 3A*). Furthermore, unlike wild type root hairs whose elongation was stimulated by auxin, *llg1* root hair defects were not mitigated by auxin and emerged root hairs in *llg1* remained shorter and substantially less sensitive to auxin stimulation (*Figure 3B*), similar to *fer-4*. Both *fer-4* and *llg1* seedlings were also defective in their epidermal cell pattern (*Figure 3C*), another auxin- and RAC/ROP-regulated property (*Wu et al., 2011*). Complementary lobes and indents along the surfaces of neighboring cells were considerably suppressed in *llg1* and *fer-4*, giving rise to more box-shaped epidermal cells clearly distinguishable from those of a wild type seedling epidermis with interdigitated cells patterned like a jig-saw puzzle. Moreover, both FER (*Yu et al., 2012*) and RAC/ROPs (*Lemichez et al., 2001*; *Zheng et al., 2002*; *Yu et al., 2012*) down-regulate ABA signaling. Like *fer-4*, *llg1-2* was similarly hypersensitive to ABA-inhibited seedling development (*Figure 4A*; *Figure 4—figure supplement 1A,B*).

RALF1 treatment of *llg1-2* and *fer-4* seedlings showed that these mutants were comparably less sensitive than wild type plants to RALF1-mediated growth responses. At the end of a 2-day treatment period, wild type seedlings were evidently shorter than their mock-treated counterparts, while treated

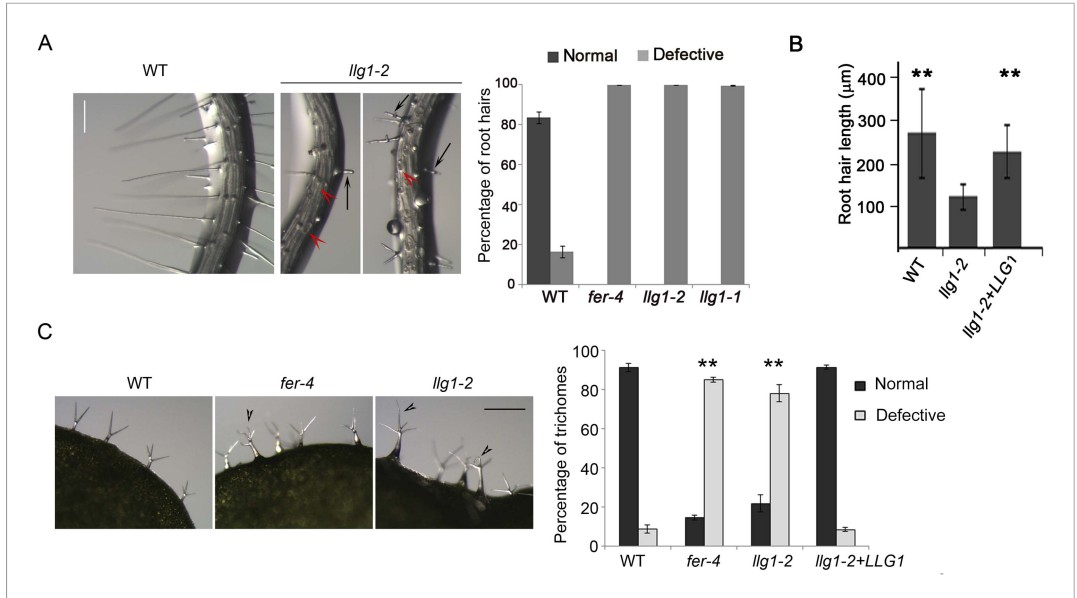

**Figure 2**. Root hair and trichome defects in *llg1* are indistinguishable from *fer*. (**A**) Root hair phenotypes (4-day-old seedlings). Defective root hairs in *llg1* include short hairs (black arrows) and those that collapsed upon emergence (red arrowheads). Histogram data: average ± SD (n≥1000 root hairs). (**B**) Root hair length comparison. Data: average root hair lengths ± SEM (n=14 four-day-old seedling roots, between 50 and 100 root hairs were measured per root). Root hair lengths for *llg1-2* were obtained from short root hairs with measurable lengths and did not include collapsed root hairs, which had no measurable lengths; thus the data shown for *llg1-2* are an under-representation of the severity of its root hair defects. (**C**) Trichome phenotypes. Arrowheads indicate abnormal trichomes (curly and/or more than three branches). Data: average ± SD (n≥400 trichomes from 3-week-old plants). **p<$10^{-2}$ to <$10^{-4}$, highly significant differences from control, as determined by the Student's *t*-test here and for the rest of the results. Scale bars: 100 µm (**A**), 0.5 mm (**C**). Additional data are shown in *Figure 2—figure supplement 1* and *Figure 2—figure supplement 2*.

The following figure supplements are available for figure 2:

**Figure supplement 1**. LLG1 complements *llg1-2*.

**Figure supplement 2**. Phenotypes in *fer-4 llg1-2* double mutant are comparable to *fer-4* and *llg1-2* single mutants.

---

*fer-4* (*Haruta et al., 2014*) and *llg1-2* seedlings as a population remained comparable in their sizes to their untreated counterparts (*Figure 4B*; *Figure 4—figure supplement 2A,B*). Since growth was not uniformly suppressed among *fer-4* and *llg1* mutants (see *Figure 2—figure supplement 2A–C*; *Figure 4—figure supplement 2*), actual root growth during the 2 days of RALF1 treatment was therefore measured to better quantify the response to RALF1. While wild type seedling growth during RALF1 treatment was significantly retarded relative to the control seedlings, RALF1-treated *fer-4* and *llg1-2* seedlings grew comparably with their mock-treated counterparts (*Figure 4C*; *Figure 4—figure supplement 2C*). Sensitivity to RALF1-regulated gene expression was also examined. When two representative RALF1-stimulated and two RALF1-suppressed genes (*Haruta et al., 2014*) were examined, their expression levels in *llg1-2* and *fer-4* were also less sensitive to the impact of RALF1 than in wild type seedlings (*Figure 4D*).

These observations together indicate that FER and LLG1/LRE share the same function in several hormone- and RAC/ROP GTPase-mediated responses and they are both required for at least a subset of FER-mediated pathways.

## LLG1 is a component of the FER-ROPGEF-RAC/ROP signaling complex

We demonstrated previously that FER interacts with RAC/ROPs in a multi-component signaling complex (*Duan et al., 2010*). Using the same protein pull-down strategy with ROP2, an Arabidopsis

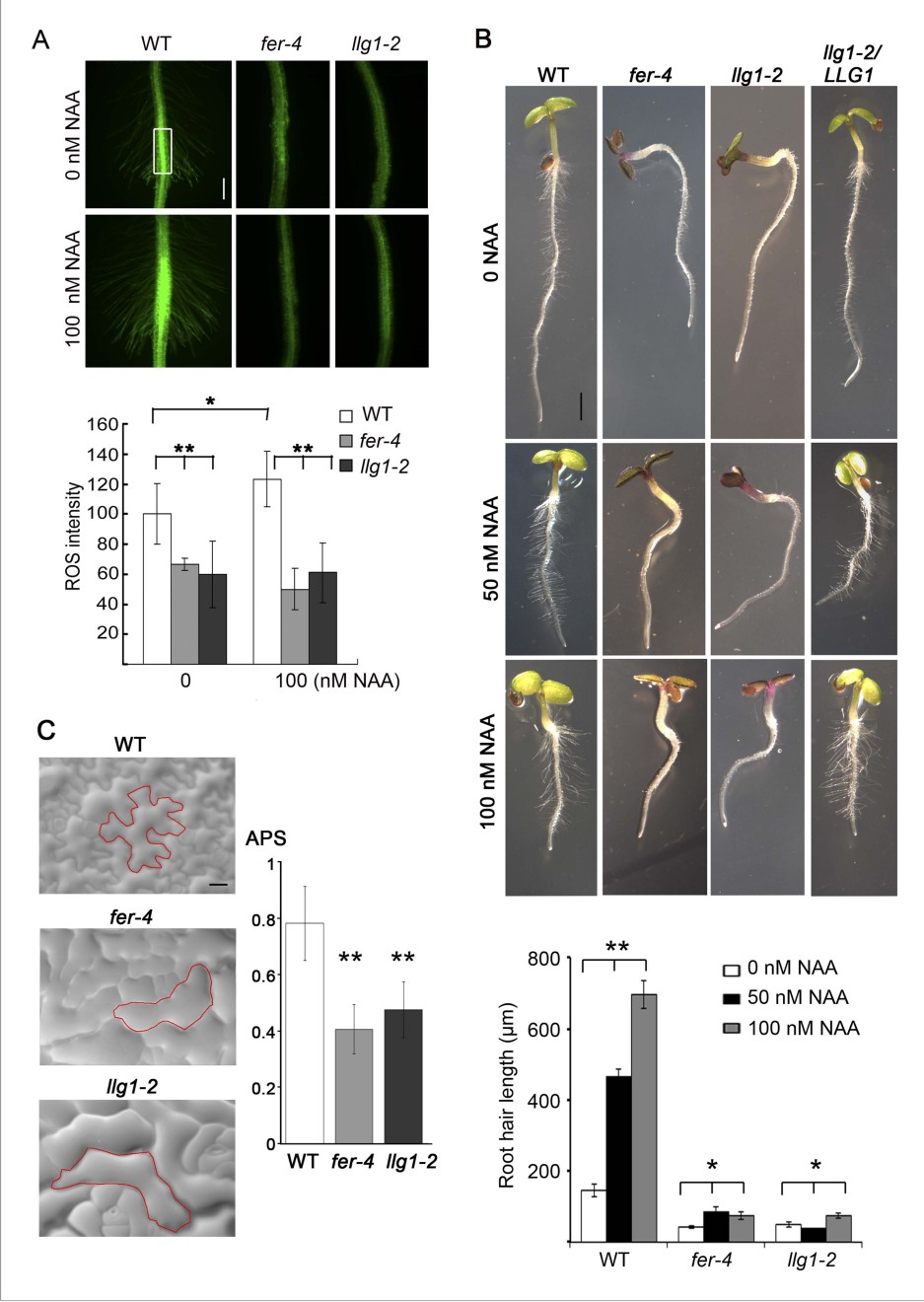

**Figure 3**. LLG1 controls auxin and RAC/ROP-regulated phenotypes. (**A**) Reactive oxygen species (ROS) accumulation and auxin-stimulated ROS production in roots. The histogram shows ROS intensity measured in identical region of interests in similar root regions (boxed area) as described (*Duan et al., 2010*). Data: averages ± SD (n=8–15 roots). The 100 nM auxin (NAA) data for wild type (WT) root were an underestimation as some auxin-stimulated ROS signals had saturated detection sensitivity. (**B**) Auxin-regulated root hair growth response. Data: averages ± SEM (n=triplicate samplings of five 4-day-old seedling roots, 100 root hairs from each root). Auxin stimulated wild type root hair lengths significantly and was dose-dependent; its effect on *llg1-2* and *fer-4* root hairs was negligible or barely reached significance. (**C**) Epidermal pavement cells from 6-day-old seedlings. Average polarity score (APS) was determined as described (*Le et al., 2006*; *Sorek et al., 2011*). Data: averages ± SD (n=20 cells). *p<0.05, **p<$10^{-2}$, significant and highly significant differences, respectively, from control. Scale bars: 100 µm (**A**); 2 mm (**B**); 50 µm (**C**).

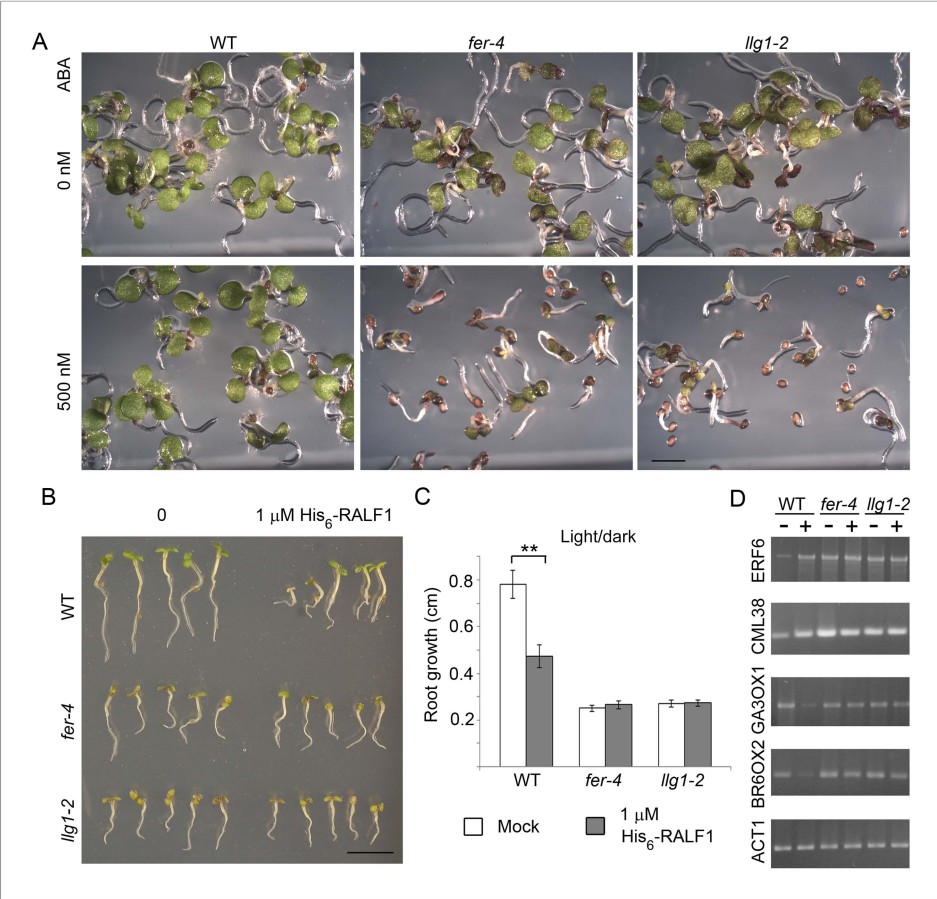

**Figure 4**. *llg1-2* and *fer-4* show similar responses to abscisic acid and RALF1. (**A**) Five-day-old seedlings with or without (ABA) treatment. (**B–D**) RALF1 treatments. (**B**) Seedlings after 2 days of mock (0) or RALF1 treatment. (**C**) Comparison of root elongation during 2 days of RALF1 treatment. Data: average $\pm$ SEM (n=3 replicate experiments); **p<$10^{-2}$, significant difference. Scale bars: 2 mm (**A**); 5 mm (**B**). (**D**) RT-PCR analysis of RALF1-regulated genes in mock (–) and 1 µM RALF1 (+) treated seedlings. Target genes analyzed were a subset of RALF-regulated genes (*Haruta et al., 2014*). ACT1: actin 1, as control; BR6OX2: brassinosteroid-6-oxidase 2; GA3OX1: gibberellin-3-oxidase 1; CML38: camoldulin-like 38; ERF6: ethylene response factor 6; WT: wild type. *Figure 4—figure supplement 1* and *Figure 4—figure supplement 2* show additional ABA and RALF1 treatment data.

The following figure supplements are available for figure 4:

**Figure supplement 1**. Abscisic acid (ABA) treatment.

**Figure supplement 2**. RALF treatment.

RAC/ROP, as bait, we observed that LLG1 was also pulled-down by ROP2 and in a guanine nucleotide-dependent manner, favored by GDP (*Figure 5A*), as was LRE (*Figure 5*; *Figure 5—figure supplement 1*). Distinct families of effectors are targeted by activated RAC/ROPs to mediate downstream pathways (*Lavy et al., 2007*; *Wu et al., 2011*). We observed that GTP-saturated ROP2 preferentially pulled-down the N-terminal fragment of the Arabidopsis RbohD-encoded NADPH oxidase (*Figure 5B*), indicating that it is also a RAC/ROP effector, as previously demonstrated for rice RAC1 and RbohB (*Wong et al., 2007*). FER was identified as a ROPGEF interacting protein (*Duan et al., 2010*) and GEF-RHO GTPase interaction is well established as has been demonstrated for ROPGEFs and RAC/ROPs in plants (*Berken et al., 2005*). Results reported here therefore imply a tetrameric FER-LLG1/LRE-ROPGEF-RAC/ROP signaling complex mediated by preferential ROPGEF binding to GDP-bound inactive RAC/ROPs (*Duan et al., 2010*), and once activated, the GTP-bound

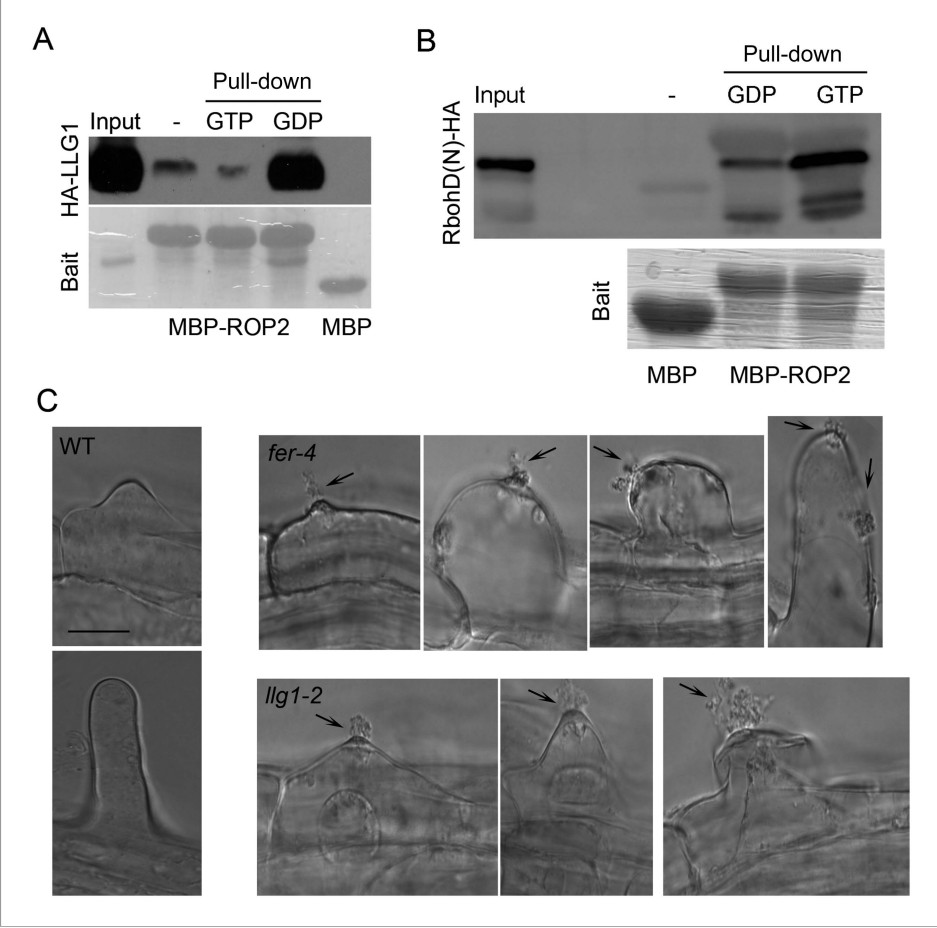

**Figure 5**. LLG1 and FER function in a RAC/ROP signaling complex and control reactive oxygen species (ROS)-mediated cell wall integrity. (**A**) ROP2-MBP pull-down of LLG1 is GDP-enhanced. *Figure 5—figure supplement 1* shows similar guanine nucleotide-dependent pull-down of LRE. (**B**) ROP2-MBP pull-down of RbohD(N) is GTP-enhanced. RbohD(N) is the N-terminal fragment (see *Supplementary file 1*) analogous to that of OsRbohB, a rice OsRac1 effector (*Wong et al., 2007*). Pulled-down proteins were detected by anti-HA antibody. Lower panels show comparable bait usage. (**C**) Emerging and elongating root hairs. Arrows indicate cytoplasmic discharge. Many *fer-4* and *llg1-2* root hairs were leaking cytoplasm (arrows) prior to collapse. A large amount of cytoplasmic mass could be seen deposited outside hairs that already appeared flaccid (*llg1-2*, right panel). WT: wild type. Scale bar: 10 μm.

The following figure supplement is available for figure 5:

**Figure supplement 1**. Guanine nucleotide-dependent pull-down of LRE by MBP-ROP2.

activated RAC/ROPs recruit NADPH oxidases to mediate downstream ROS-dependent processes. Emerging root hairs in *fer-4* and *llg1* seedlings began to extrude cytoplasm prior to their collapse (*Figure 5C*), consistent with a weakening cell wall as a result of reduced ROS production (*Foreman et al., 2003*; *Carol and Dolan, 2006*; *Swanson and Gilroy, 2010*) that ultimately led to their collapse.

## Loss of LLG1 and LRE functions suppresses FER localization to the cell membrane

Functional interactions between FER and LLG1 occur beyond acting as components of the RAC/ROP signaling apparatus. In wild type vegetative organs, FER promoter-expressed FER-GFP localized to the cell membrane and intracellular signal was negligible (*Figure 6A*; *Escobar-Restrepo et al., 2007*; *Duan et al., 2010*). On the other hand, the intracellular FER-GFP signal was considerably more pronounced in *llg1* hypocotyl (*Figure 6B*) and roots (*Figure 6*; *Figure 6—figure supplement 1*) than

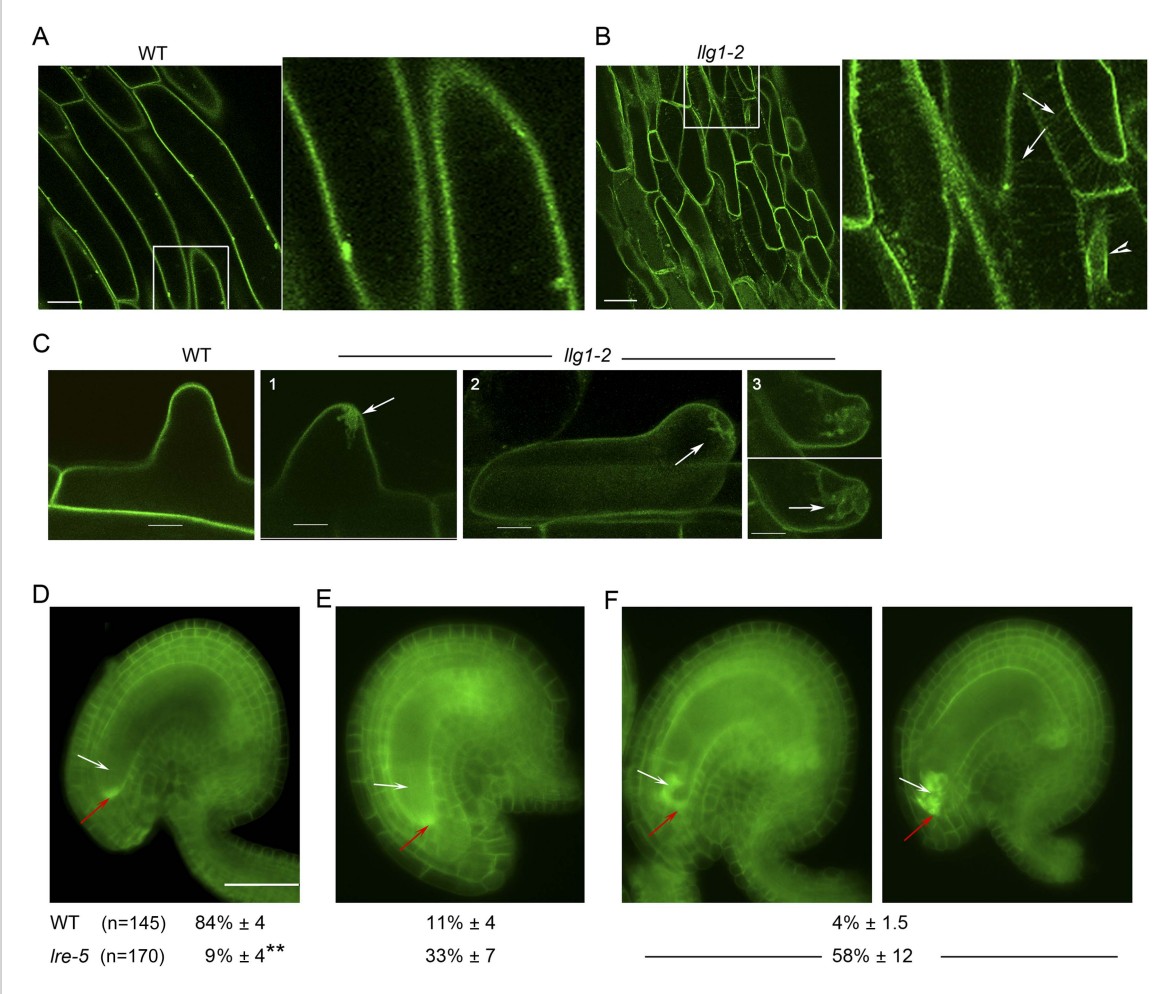

**Figure 6**. LLG1 and LRE are important for FER cell membrane localization. (**A**, **B**) *pFER*-expressed FER-GFP localization in 5-day-old wild type (WT) and *llg1-2* seedling hypocotyl. Boxed areas are magnified (4x) to highlight FER-GFP-labeled reticulate arrays (arrows) and perinuclear region (arrowhead) in *llg1-2*, which are absent from wild type. ***Figure 6—figure supplement 1A,B*** shows root samples. (**C**) Emerging root hairs. Arrows indicate intracellular FER-GFP. ***Figure 6—figure supplement 1C,D*** shows colocalization with ER-Tracker. (**D–F**) FER-GFP localization in ovules. (**D**) Normal FER-GFP localization in ovules, predominantly seen in wild type ovules. (**E**) *lre-5* ovules with moderate FER-GFP mislocalization; filiform apparatus and synergid cell signal quantification is shown in ***Figure 6—figure supplement 2***. (**F**) *lre-5* ovules with severe FER-GFP mislocalization; the signal retained in the synergid cells sometimes appeared patchy (right image; see also ***Figure 6—figure supplement 2A***). (**A–C**) Single optical sections; (**D–F**) wide-field images. Scale bars: 25 µm (**A**, **B**); 5 µm (**C**); 50 µm (**D**).

The following figure supplements are available for figure 6:

**Figure supplement 1**. FER-GFP localization in *llg1-2* roots.

**Figure supplement 2**. Distribution of filiform apparatus and synergid cell (SC) cytoplasmic FER-GFP signal in wild type and *lre-5* ovules.

in wild type tissues, often appearing in reticulate arrays and perinuclear regions, patterns reminiscent of the ER. Intracellular FER-GFP signal was prevalent in emerging root hairs prior to or during their rupture (***Figure 6C***), and colocalized with ER-Tracker-stained membrane patches (***Figure 6—figure supplement 1C,D***).

In the ovules, FER-GFP appears most prominently at the filiform apparatus (***Figure 6D***; ***Escobar-Restrepo et al., 2007***; ***Duan et al., 2014***), a synergid cell membrane-enriched cell wall region at the entrance to the female gametophyte (***Kasahara et al., 2005***). The concentrated localization of FER-GFP at the filiform apparatus was highly compromised in *lre* ovules. A significantly higher percentage

of mutant ovules relative to wild type showed female gametophyte FER-GFP either totally retained in the synergid cell cytoplasm (*Figure 6F*) or distributed between the filiform apparatus and inside the synergids (*Figure 6E*; *Figure 6—figure supplement 2B*). FER-GFP retained inside the synergids sometimes appeared as cytoplasmic patches (*Figure 6F*; *Figure 6—figure supplement 2A*), reflecting a dense synergid cell endomembrane system (see, for example, *Kasahara et al., 2005*) that supports secretion of cell wall materials to construct the filiform apparatus.

## Loss of LLG1 induces retention of FER in the ER

To further understand how loss of LLG1 affects FER-GFP localization, we transformed protoplasts derived from wild type and *llg1* plants, where the localization of 35S promoter-expressed FER-GFP (*35S::FER-GFP*) could be clearly discerned and dose-dependence on the introduced transgenes could be quantitatively assessed. When transformed using the same amount of input FER-GFP DNA, *llg1-2*-derived protoplasts showed considerably stronger intracellular signals than their counterpart wild type cells (*Figure 7*; *Figure 7—figure supplement 1* and *Figure 7—figure supplement 2*), consistent with intracellular FER-GFP signal being prominent in *llg1* plants (*Figure 6*). The cytoplasmic FER-GFP signal in *llg1* protoplasts often appeared in reticulate structures, reminiscent of the ER, and colocalized with a co-expressed ER marker (RFP-ER) (*Figure 7C,D*; *Figure 7—figure supplement 2B*,

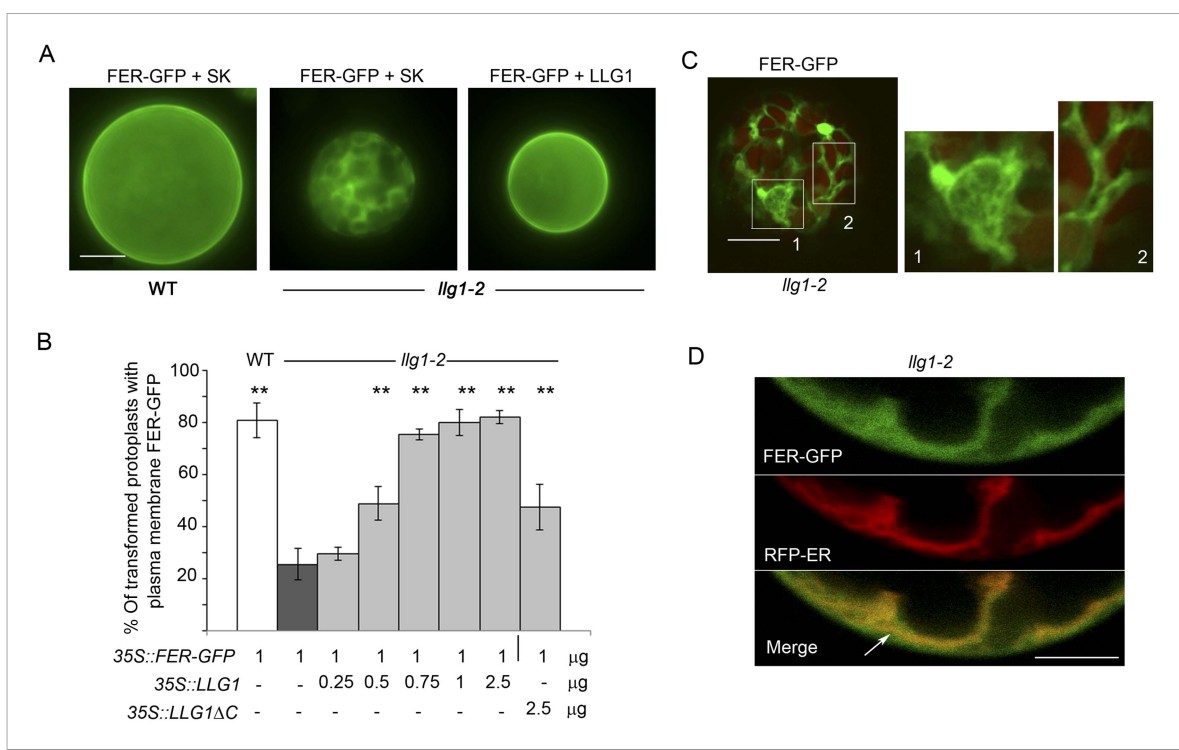

**Figure 7**. Loss of LLG1 induces endoplasmic reticulum (ER) retention of FER-GFP. (**A**) FER-GFP localization in transiently transformed wild type (WT) and *llg1* protoplasts. SK, vector DNA. (**B**) FER-GFP localization response in *llg1-2* protoplasts to co-transforming LLG1 or LLG1ΔC DNA. Data: average ± SD (n=triplicate samplings of 100 transformed protoplasts in each sample). **p<10⁻², both significantly different from FER-GFP transformed *llg1-2* (second data bar). Tagged LLG1s were comparably active (see *Figure 7—figure supplement 1A*). (**C**) FER-GFP retained in reticulate structures (magnified boxes 1, 2) in *llg1-2* protoplasts; red indicates chlorophyll autofluorescence. (**D**) Localization of FER-GFP with co-expressed RFP-ER in *llg1-2* cells; arrow indicates cell membrane–cortical ER boundary. See also *Figure 7—figure supplement 2B–D*. (**A**) Wide-field images; (**C**, **D**) single confocal sections. Scale bars: 10 μm.

The following figure supplements are available for figure 7:

**Figure supplement 1**. *35S::FER-GFP* transformed wild type (**A**) and *llg1-2* (**B**) protoplasts.

**Figure supplement 2**. Efficient cell membrane localization of FER-GFP depends on LLG1.

*C*). Cell surface localization of FER-GFP was not obliterated from *llg1* protoplasts, but could clearly be resolved from its ER locations in cells where cortical ER was prominent (*Figure 7D*). Co-transfection of *llg1* protoplasts by LLG1, epitope- or fluorescent protein-tagged LLG1 all restored cell membrane localization of FER-GFP (*Figure 7A,B*; *Figure 7—figure supplement 2A,D*). On the other hand, expression of LLG1ΔC, deleted of its C-terminal signature sequence for GPI-anchor modification (*Kinoshita, 2014*), was considerably less effective in restoring FER-GFP to the plasma membrane of *llg1* cells (*Figure 7B*). Together with observations in *llg1* seedlings (*Figure 6*), these results indicate that FER depends on GPI-anchored LRE and LLG1 for efficient cell membrane location.

## FER interacts with LLG1 on the cell membrane and in the ER

Results thus far support the notion that LLG1 and LRE are critical for the biological functions of FER by facilitating localization of FER to the cell membrane (*Figures 6, 7*) where they also associate with the RAC/ROP complex (*Figure 5*) to mediate downstream processes (*Figures 3, 4*). To understand how LLG1 and LRE could attain these two properties, we explored whether they might interact directly with FER (*Figures 8, 9*). Protein–protein interaction assays showed that protoplast-expressed FER was pulled-down by LLG1 and LRE in vitro (*Figure 8B*) and FER co-immunoprecipitated with LLG1 that was co-expressed in protoplasts (*Figure 8C*), suggesting direct interactions between these proteins in plant cells. Moreover, protoplasts-coexpressed FER-GFP and LLG1-HA were co-pulled down by MBP-RALF1 (*Figure 8D*), indicative of RALF1 interaction with the FER-LLG1 complex.

Bimolecular fluorescence complementation (BiFC) assays (*Ohad and Yalovsky, 2010*) were used to further examine FER-LLG1 interaction in plant cells (*Figure 9*; *Figure 9—figure supplement 1*). LLG1 and a truncated version of FER deleted of its kinase domain (FERΔK) (*Figure 8A*), which could be

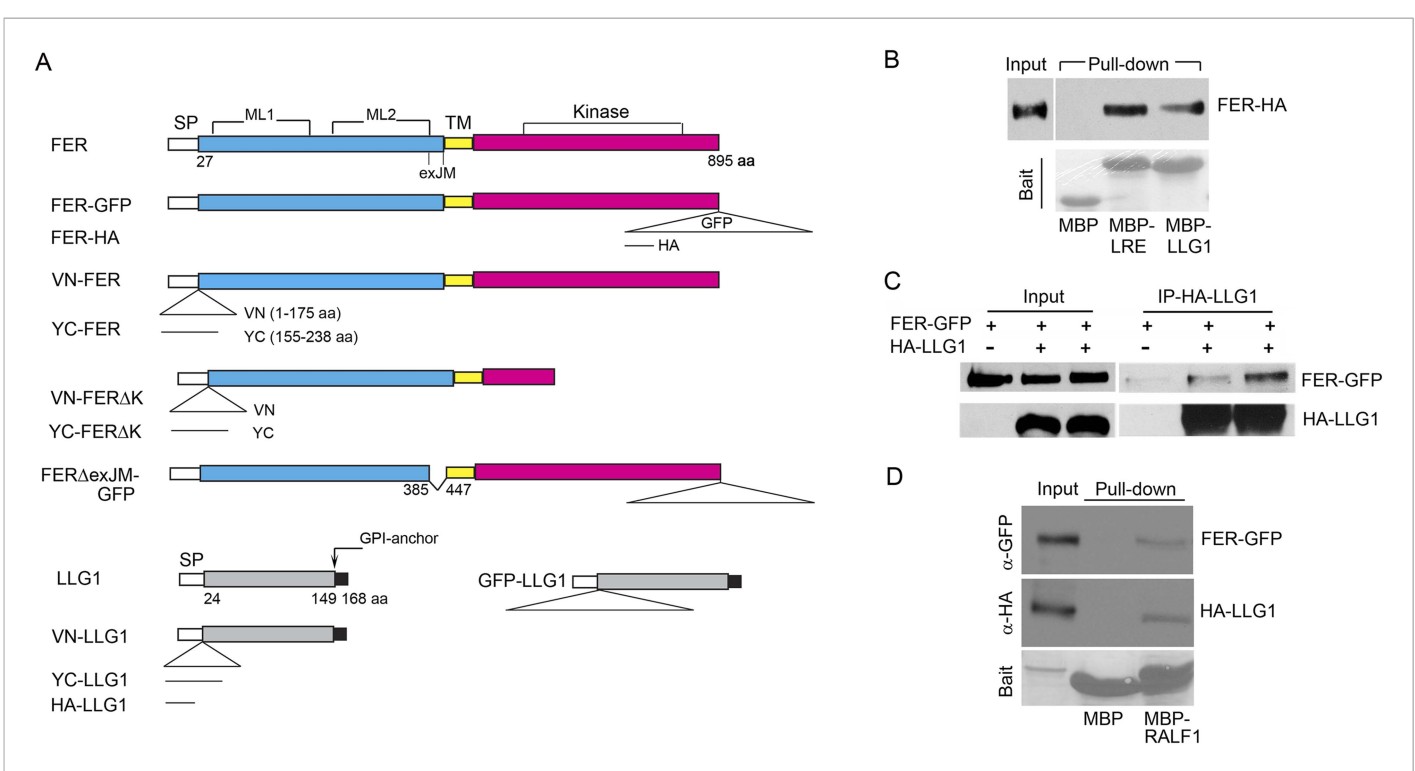

**Figure 8**. LLG1 and LRE interact with FER. (**A**) Domain maps for FER, LLG1, and key constructs used for protein–protein interaction studies. *Supplementary file 1* shows a full list of constructs used. SP, TM, ML1,2, and exJM are signal peptide, transmembrane, malectin-like, and extracellular juxtamembrane domains, respectively. ΔexJM, exJM deletion. Triangles indicate insertion locations for the various tags used. (**B**) *Escherichia coli* produced LLG1-MBP and LRE-MBP pull-down of protoplasts-expressed FER-HA, detected by anti-HA antibody (upper panel). The lower panel shows the comparable amount of bait proteins used. (**C**) Co-immunoprecipitation of protoplasts-expressed FER-GFP and HA-LLG1 by anti-HA; FER-GFP was detected by anti-GFP. (**D**) MBP-RALF1 pull-down of FER-GFP and HA-LLG1 co-expressed in transformed protoplasts. Anti-HA or anti-GFP antibodies were used in these blots. Bait panels were Ponçeau S-stained blots showing comparable amounts of bait usage.

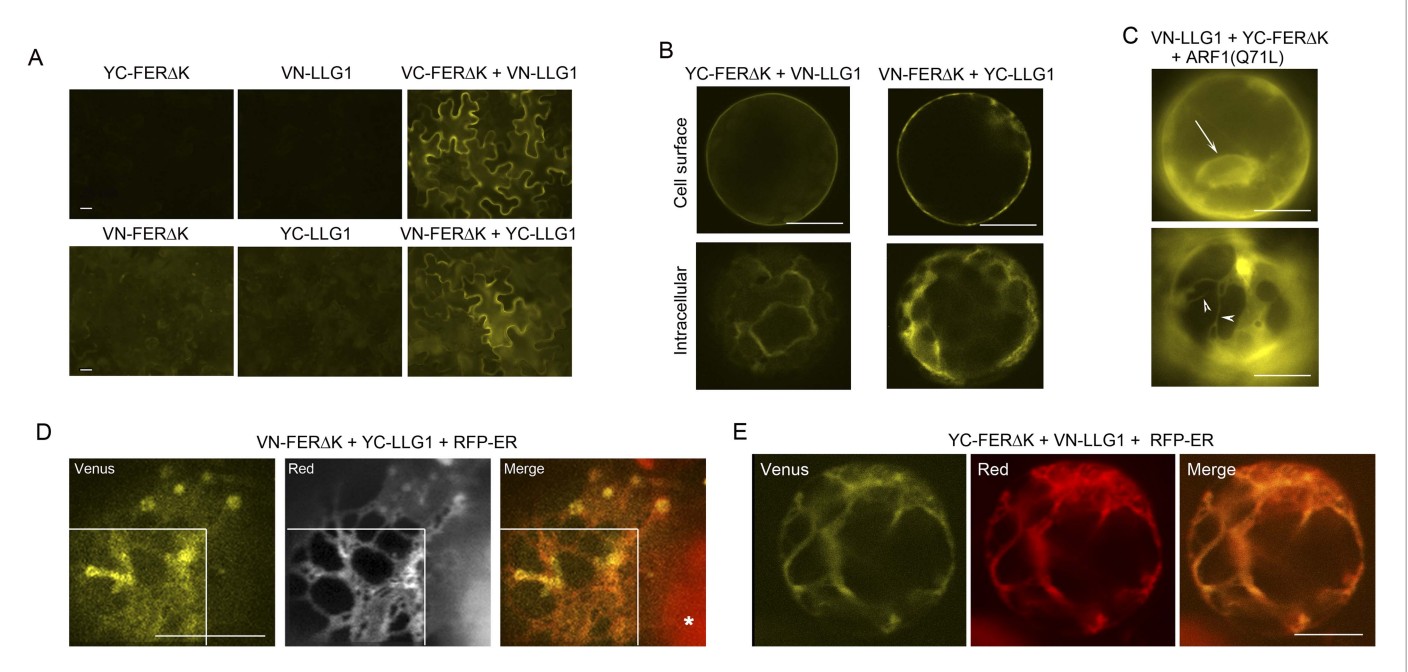

**Figure 9**. BiFC assays show LLG1-FER interaction on the cell membrane and in the endoplasmic reticulum (ER). Transgenes used are as indicated. (**A**) Agroinfiltration-transfected tobacco leaf epidermal cells. *Figure 9—figure supplement 1A* shows FER-GFP and GFP-LLG1 localization when expressed alone. (**B**–**E**) Transformed Arabidopsis protoplasts. (**B**) Protoplasts with cell membrane-located reconstituted Venus signal (upper panel) and pronounced intracellular signal (lower panel) were observed on comparable levels (n>3 replicate assays, with ~50 BiFC positive cells each). *Figure 9—figure supplement 1* shows GFP-LLG1 localization when expressed alone, single vector controls, and reconstituted Venus signal in cells expressing the full-length FER as one of the split Venus pair. (**C**) The majority of protoplasts (>90%, n=triplicate samplings, ~50 BIFC positive cells each) co-transformed by ARF1(Q71L) and the split Venus pair showed strong intracellular Venus signals that resembled the ER. Arrow indicates perinuclear; arrowheads indicate dynamic reticulate structure (see *Videos 1, 2*). (**D**, **E**) Colocalization of the split Venus signal with the ER marker, RFP-ER. Boxed areas in (**D**) were contrast-enhanced (equally) to highlight the signals. *, chlorophyll fluorescence. (**A**, **C**) Wide-field, (**B**, **D**, **E**) confocal images. Scale bars: 25 µm (**A**), 10 µm (**B**–**E**).

The following figure supplement is available for figure 9:

**Figure supplement 1**. Additional controls and supportive data for BiFC studies.

expressed at higher levels than full-length FER, were reciprocally fused to the N- and C-terminal halves (VN, YC, respectively) of Venus. These split Venus halves were also fused behind the FER signal peptide for targeting to the secretory pathway. Co-expression of cognate pairs of these split Venus in agroinfiltration transformed tobacco leaf epidermis reconstituted Venus yellow fluorescence along the cell surface (*Figure 9A,B*), reflecting FER-LLG1 interaction on the leaf cell membrane, and consistent with FER-GFP and GFP-LLG1 being predominantly located to the cell surface when expressed alone in similar assays (*Figure 9—figure supplement 1A*).

In transformed Arabidopsis protoplasts for BiFC analysis, typically about half of the BiFC positive cells displayed prominent cell membrane-localized signal with little notable intracellular signals (*Figure 9B*, upper panel). The other half harbored prominent Venus signal inside the cell, some appearing in reticulate structures, along with signal on the cell surface or cell cortex (*Figure 9B*, lower panel; *Figure 9—figure supplement 1C*). When the split Venus pairs were co-expressed with RFP-ER, reconstituted intracellular yellow fluorescence colocalized with the ER marker (*Figure 9D,E*), reflecting FER-LLG1 interaction in the ER. The notable BiFC signal within the ER most probably had resulted from retarded trafficking of the interacting proteins out of this compartment. An Arabidopsis ARF1 mutation, ARF1(Q71L), has been shown to block ER to Golgi trafficking and trap secretory proteins in the ER (*Cai et al., 2011*). When co-expressed with the split Venus pair of FERΔK and LLG1, ARF1 (Q71L) dramatically increased the number of BiFC-positive cells (to >90%) showing intracellular yellow

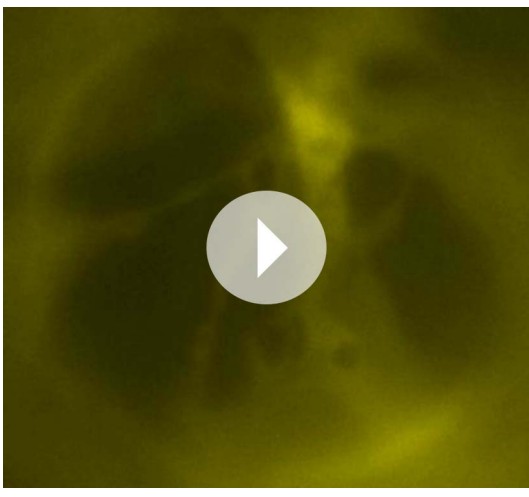

**Video 1.** Time series showing the dynamics of intracellular LLG1-FERΔK BiFC signal, typical of the endoplasmic reticulum (ER), in the protoplast shown in *Figure 9C*, lower, which was co-transformed by the split Venus pair and ARF1(Q71L) as indicated on the figure. Time series shows 89 images in about real-time.

fluorescence. The reconstituted Venus signal often appeared perinuclear and in highly dynamic reticulate structures, typical of the ER (*Figure 9C*; *Videos 1, 2*). These observations further support FER-LLG1 interaction occurred already in the ER where these proteins would first encounter each other early in the secretory process.

## LLG1 binds to the extracellular juxtamembrane region of FER

Having determined that LLG1 is crucial for FER localization to the cell membrane and that FER and LLG1 interact physically (*Figures 6–9*), we sought further evidence for these interactions and their biological significance. We observed that FERΔexJM, FER deleted of the extracellular juxtamembrane region (exJM, amino acid residues 365–447) (*Figure 8A*) was less efficiently pulled-down by LLG1 (*Figure 10A*), while the 83 amino acid exJM fragment was adequate to pull-down LLG1 (*Figure 10B*). Yeast two-hybrid assays also indicated that exJM interacted with LLG1 and LRE (*Figure 10C*; *Figure 10—figure supplement 1*). Furthermore, when *pFER::FERΔexJM-GFP* was transformed into Arabidopsis, FERΔexJM-GFP showed prominent intracellular localization in structures reminiscent of the ER (*Figure 10D,E*; *Figure 10—figure supplement 2A*). These intracellular FERΔexJM-GFP signals also colocalized with the ER-tracker dye (*Figure 10E*) and with RFP-ER when co-expressed in protoplasts

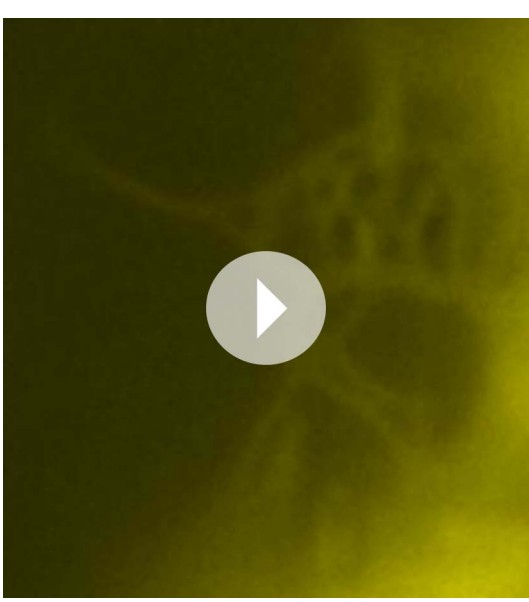

**Video 2.** Time series showing the dynamics of intracellular LLG1-FERΔK BiFC signal in a protoplast that co-transformed by the split Venus pair and ARF1 (Q71L) in which the ER-reticulate structure is apparent. Time series shows 200 images, about 3× accelerated from acquisition time.

(*Figure 10—figure supplement 2B*). These observations together indicate that LLG1 and LRE target the exJM in FER for binding and that FER depends on these interactions for proper delivery to the cell membrane. They also imply that the phenotypes in *llg1* and *lre*, together spanning the developmental spectrum of *fer* phenotypes, could have largely resulted from diminished FER localization to its proper functional location in the cell membrane, thus mimicking loss of FER.

## Discussion

### Differentially expressed LLG1 and LRE as functional partners to diversify the biological roles of FER

Results reported here demonstrate that related GPI-APs LLG1 and LRE are functional counterparts that physically interact with FER and essential for its cell surface signaling capacity. Partnering with and relying on this differentially expressed protein pair clearly would provide versatility for the almost constitutively expressed FER to control when and where signaling can be deployed. Moreover, modulation of LLG1 and LRE expression and their post-translational modifications, which involve numerous GPI-anchor

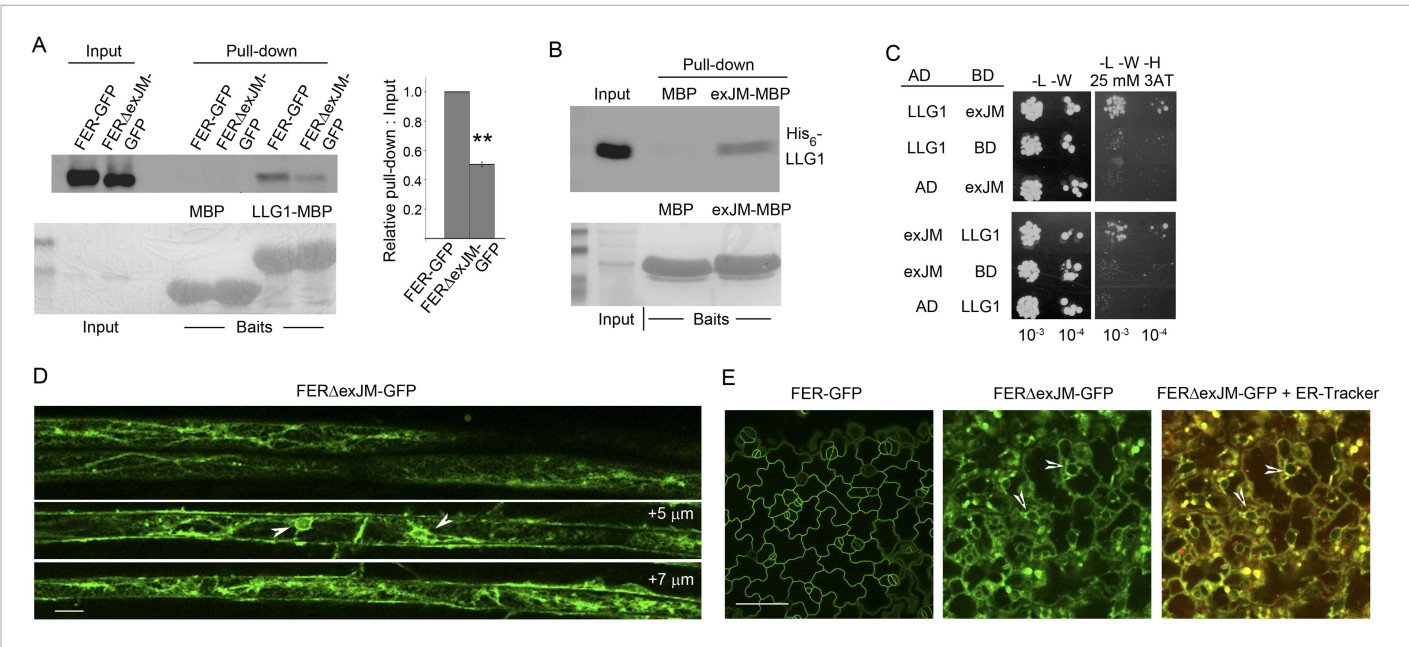

**Figure 10**. LLG1 interacts with exJM and underlies FER localization to the cell membrane. (**A**) LLG1-MBP pull-down of protoplasts-expressed FER-GFP and FERΔexJM-GFP. Histogram shows relative pull-down efficiency (pull-down signal: input signal); n=3 independent experiments, pull-down efficiency for FER-GFP in each experiment was normalized to 1; **p<10$^{-2}$. (**B**) exJM fragment pull-down of His$_6$-tagged LLG1; both were expressed in *Escherichia coli*. (**C**) Yeast two-hybrid assays between exJM and LLG1 (amino acids 24–149, *Figure 8A*) on 3AT selection. *Figure 10—figure supplement 1* shows β-galactosidase assays for LLG1-exJM interaction and exJM-LRE (amino acids 21–138) interaction. (**D**, **E**) FERΔexJM-GFP localization in root (**D**) and cotyledon epidermal cells (counter-stained by ER-Tracker Red) (**E**) of *pFER::FERΔexJM-GFP* transformed Arabidopsis seedlings (4 days old). (**D**) Optical sections from a Z-stack (see *Figure 10—figure supplement 2A*) selected to highlight the endoplasmic reticulum (ER)-reminiscent reticulate and perinuclear (arrowheads) labeling pattern; +5 µm and +7 µm indicate distances from the image shown at the top. Scale bars: 20 µm (**D**), 25 µm (**E**).

The following figure supplements are available for figure 10:

**Figure supplement 1**. Yeast two-hybrid analysis of FERexJM-LLG1/LRE interaction.

**Figure supplement 2**. FERΔexJM-GFP localization.

biosynthetic and restructuring steps (*Fujita and Kinoshita, 2012*; *Kinoshita et al., 2013*; *Cheung et al., 2014*), would provide many opportunities for FER signaling activity to be fine-tuned to meet a broad range of developmental needs and environmental challenges. Given the spectrum of phenomena already known to be regulated by FER (*Wolf and Hofte, 2014*), LLG1 and LRE could be of core importance to how FER treads among several signaling pathways to mediate growth and developmental processes regulated by multiple hormones, reproduction, and defense, each potentially controlled by distinct signals. FER is a member of a family of 17 related Arabidopsis receptor kinases, which are also conserved in other plant species (*Hematy and Hofte, 2008*); LRE-like proteins are also present in other plants (*Tsukamoto et al., 2010*). It is conceivable that pairing with their cognate LRE-like proteins also underlies the signaling functions of other FER-related receptor kinases. GPI-APs, including, for example, the COBRA family proteins (*Schindelman et al., 2001*; *Li et al., 2013*) and arabinogalactan proteins (*Demesa-Arevalo and Vielle-Calzada, 2013*), play crucial roles in plant growth and reproduction. Demonstrating that FER depends on LLG1 and LRE for efficient localization to the cell membrane (*Figures 6, 7*) elucidates not only a strategy whereby FER acquires its cell surface signaling capacity but also a mechanism with which GPI-APs might enable the signaling capacity of a broader set of cell surface receptors.

## LLG1 and LRE as 'chaperones' for FER delivery to the cell membrane

The FER-LLG1/LRE partnership also underscores the importance and the versatility of GPI-APs as regulators for cell signaling activities. GPI-APs are secreted to the outer leaflet of the cell membrane.

LLG1 and LRE binding to the exJM domain of FER (*Figure 9*) is consistent with their being tethered to the cell membrane via the GPI anchor, thus facilitating interactions with FER immediately along the outer cell surface. GPI anchors are assembled and transferred to proteins destined for lipid modification in the ER. GPI-APs are known to undergo structural remodeling in the ER for efficient trafficking to the Golgi in vesicles that are distinct from those for other secretory proteins, already marking them for delivery to specialized sphingolipids- and cholesterols-enriched cell membrane microdomains (*Fujita and Kinoshita, 2012*; *Kinoshita et al., 2013*). With its signal peptide, FER could be secreted by default to the cell membrane. That FER and LLG1 already interact in the ER (*Figure 9*) and this interaction underlies efficient FER localization to the cell membrane (*Figures 6, 7 and 10*) are consistent with at least a fraction of FER exiting the ER as a FER-LLG1/LRE complex (*Figure 11*). That FER and LLG1/LRE remain associated with the RAC/ROP signaling complex along the cell membrane

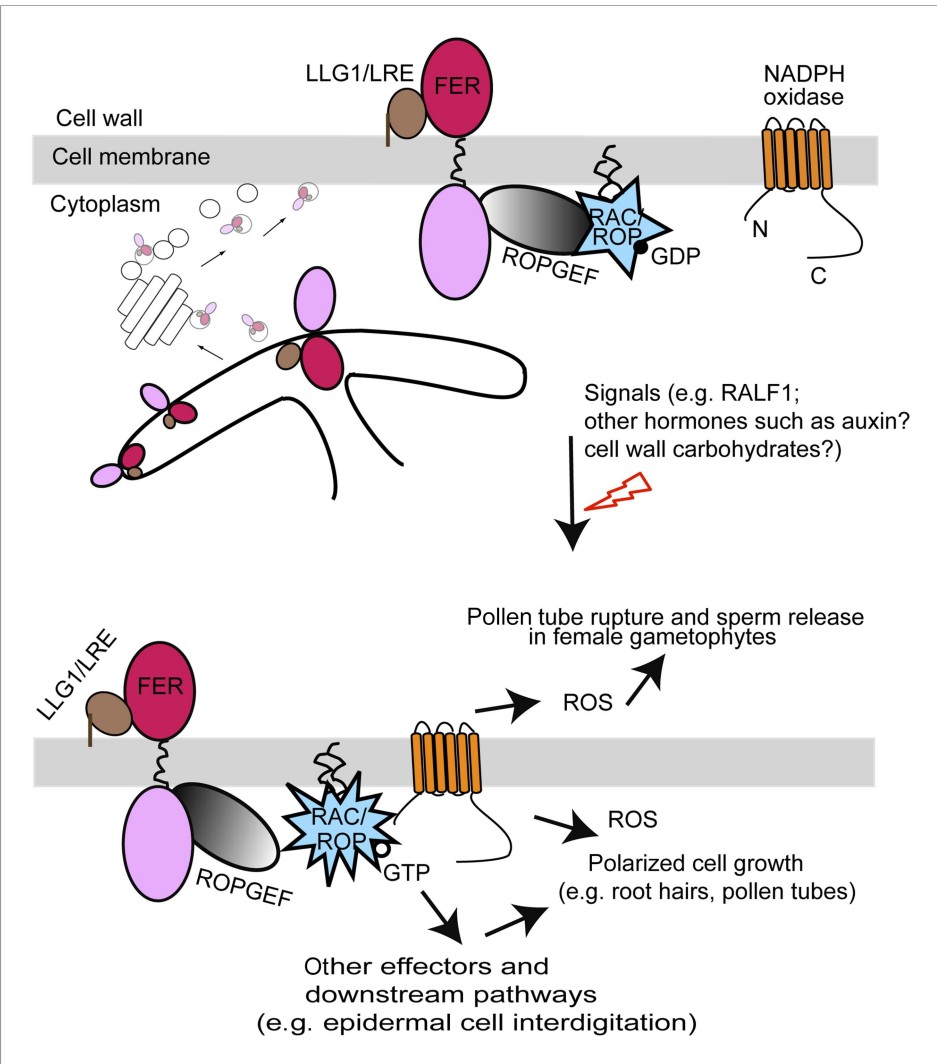

**Figure 11**. A model for FER-LLG1/LRE interaction and mediated RAC/ROP signaling. FER-LLG1/LRE are shown interacting in the endoplasmic reticulum (ER) and on the cell membrane where they exist as components of a RAC/ROP signaling complex. FER-ROPGEF (*Duan et al., 2010*) and ROPGEF-RAC/ROP (*Berken et al., 2005*) interactions were previously established. RAC/ROPs pull-down FER (*Duan et al., 2010*), LRE, and LLG1 (*Figure 5A*) in a GDP-dependent manner. Upon signal activation, activated GTP-bound RAC/ROPs target effectors, such as NADPH oxidase (*Wong et al., 2007*) (*Figure 5B*), to mediate downstream processes, including ROS-mediated polarized root hair growth and pollen tube rupture (*Duan et al., 2010*, *2014*). FER-LLG1 interaction in the ER and FER dependence on LLG1 and LRE to localize to the cell membrane suggest co-delivery of FER-LLG1/LRE complex to the cell membrane. ROS, reactive oxygen species.

(*Figures 5, 9*) also provides additional support for FER-LLG1/LRE being transported from the ER to the cell membrane as a complex. In complexing with FER in the ER and 'chaperoning' its delivery to the cell membrane, LLG1 and LRE ensure delivery of FER to GPI-AP-destined micro membrane environments for its proper functional location and assembly of the RAC/ROP signaling apparatus (*Figure 11*). In the remaining part of the FER-ROPGEF-RAC/ROP signaling complex (*Figure 5*) and being required for several RAC/ROP regulated processes (*Figure 4*; *Duan et al., 2014*), LLG1 and LRE apparently also function as an integral component of the FER signal reception apparatus. They might directly participate in signal perception by FER and/or regulate how FER interacts with various possible binding targets, such as RALF1 or other hormones and cell wall carbohydrates. They also conceivably provide a landmark for the assembly of the FER signaling apparatus and regulation of its activity such as by maintaining its stability or inducing its recycling from the cell membrane, roles known to be played by GPI-APs (*Lingwood and Simons, 2010*; *Fujita and Kinoshita, 2012*; *Yu et al., 2013*). To what extent the LRE family protein controls the delivery of receptor kinases as a class or more specifically those closely related to FER remains to be determined.

### LLG1 and LRE as co-receptors for FER signaling

Results presented here also provide evidence for the notion that LLG1/LRE acts as a co-receptor to mediate at least those FER-regulated processes examined here. On the phenotypic level, growth and developmental defects in *llg1* (*Figures 1, 2*) and reproductive defects in *lre* (*Capron et al., 2008*; *Tsukamoto et al., 2010*) together span the spectrum of vegetative and reproductive phenotypes in *fer* mutants. The non-additive phenotype of *fer-4 llg1-2* double mutant relative to each of its parent single mutants (*Figure 2—figure supplement 2*) provides further support that FER and LLG1 function in and both are required for the same pathways. Biochemically, both LLG1 and LRE physically interact with FER on the cell membrane (*Figure 9*) and exist as components of the FER-ROPGEF-RAC/ROP-NADPH oxidase signaling pathway (*Figure 5*), consistent with their serving with FER as signal mediators on the cell surface to RAC/ROPs. This is further supported biologically by loss of LLG1 or LRE inducing the same signaling defects as in *fer* null mutants, including processes regulated by auxin, ABA, RALF1, and ROS (*Figures 3, 4*; *Duan et al., 2014*). Besides LLG1/LRE, RALF1 is thus far the only other molecule reported to interact with the FER extracellular domain, although the precise RALF1 target site on FER remains unknown (*Haruta et al., 2014*). That RALF1 interacts with co-expressed FER and LLG1 (*Figure 8D*) and that both FER and LLG1 are needed to mediate RALF1 signaled responses (*Figure 4B–D*) indicate that the FER-LLG1 complex indeed has the capacity to serve as a co-receptor for this first known ligand of FER. Given the already known participation of FER in multiple hormone and defense signaling pathways, the FER-LLG1/LRE complex could equally be a surface co-regulator for multiple signals. Moreover, FER is also broadly speculated to interact with cell wall carbohydrates by virtue of its extracellular homology with the disaccharide-binding malectin (*Kessler et al., 2010*; *Cheung and Wu, 2011*; *Lindner et al., 2012*; *Wolf and Hofte, 2014*). Therefore, the extracellular interactions engaged by FER are likely to be complex and influenced by multiple factors whose presence fluctuates depending on cellular and environmental conditions. The FER-LLG1/LRE partnership discovered here laid the ground work towards a more comprehensive understanding of how FER attains its multiple biological role; this will however require first deciphering other biochemical interactions maintained by FER.

## Materials and methods

### Plant growth and transformation

Plant growth followed previously described conditions (*Duan et al., 2010*). Tissue culture-grown plants were maintained on B5 medium supplemented with 1% sucrose and solidified by 0.7% agar. Seeds were cold-treated at 4°C for 2 days before being transferred to 22°C for germination and growth under 16/8 hr light/dark cycles, or in total darkness for dark-grown seedlings. For growth to maturity, seeds were either sown directly on soil, or 10-day-old tissue culture-grown seedlings were transferred to soil, and maintained in a growth chamber at 20–22°C under 16/8 hr light/dark cycles. *Arabidopsis thaliana* Col-0 was used as control for *llg1-2* (SALK_086036) and *llg1-1* (SAIL_47_G04). Both *llg1* mutants behaved similarly throughout growth and development and did not display discernable reproductive defects. Homozygous *fer-4* (*Duan et al., 2010*, *2014*) and *lre-5* (*Tsukamoto et al., 2010*) were as previously described. Double *fer-4 llg1-2* was generated by a genetic cross.

RALF1-regulated growth used *Escherichia coli*-produced His$_6$-RALF1 and followed previously described conditions (*Bergonci et al., 2014*; *Haruta et al., 2014*). Growth for RALF1 treatment for RT-PCR analysis followed *Haruta et al. (2014)*.

Arabidopsis was transformed by floral dip (*Clough and Bent, 1998*). Transient transformation assays were carried out by agroinfiltration (*Batoko et al., 2000*) of *Nicotiana tabacum* var SR1 grown at ~25°C in a growth room. A wound was made in the abaxial epidermis and about 1 ml of bacteria (at 0.1–0.4 OD$_{600}$) was injected into these spots using a 1 ml syringe. Transient transfection of Arabidopsis protoplasts from 4-week-old soil-grown wild type and *llg1-2* plants, and of tissue culture-grown wild type Arabidopsis protoplasts followed procedures in *Yoo et al. (2007)* and *Duan et al. (2010)*, respectively. Unless otherwise indicated, DNA amounts used for protoplast transfection were: 1 μg of *pFER::FER-GFP*; varying amounts of *35S::LLG1* or *35S::LLG1* derivatives (indicated in figures); 1–2 μg of the ER marker *35S::RFP-ER* (*Sinclair et al., 2009*); 4 μg of each split Venus half (*Kodama and Hu, 2012*) and 5 μg of *35S::ARF1(Q71L)* (*Cai et al., 2011*). Empty vector (Bluescript vector SK) DNA was used to equalize the amount of DNA used in comparative assays.

## Molecular and histochemical analyses

All recombinant DNA procedures followed standard and PCR-based methodology. A list of constructs is shown in *Supplementary file 1*; domain maps for some are shown in *Figure 8*. Plant genomic DNA was used for PCR analysis of T-DNA inserts in transformed plants. RNA for expression analysis by RT-PCR was isolated from 10-day- old seedlings following the manufacture's protocol (PrepEase RNA isolation kit; USB/Affymetrix, Santa Clara, CA). Histochemical staining for GUS activity followed the standard procedure (*Jefferson, 1987*). Primers for RT-PCR of RALF1-regulated genes are: (1) BR6OX2: forward, GAG ACA TCA AGA TTG GCA ACG; reverse, GTA AGG TGA ACA CTT AAG ATGG; (2) GA3OX-1: forward, CAA GTA TTT CGC GAT GAT CTT GG; reverse, G ATA CTC TTT CCA TGT CAC CG; (3) CML38: forward, ATG AAG AAT AAT ACT CAA CCT C; reverse, GCG CAT CAT AAG AGC AAA CTC; (4) ERF6: forward, ATG GCT ACA CCA AAC GAA GTA TC; reverse, AAC AAC GGT CAA TTG TGG ATA ACC.

## Plant phenotype analyses

Plant phenotype and data analyses mostly followed *Duan et al. (2010)*. Root hairs located between 1.5 and 3.5 mm from the primary root tip of 4-day-old seedlings were examined. For auxin treatments, 1-naphthaleneacetic acid (NAA) was added at concentrations indicated in the figures. ABA treatment followed that in *Yu et al. (2012)*; hormone was added directly to seed germination plates. For RALF1 treatments, *E. coli*-produced His$_6$-RALF1 was purified according to *Morato do Canto et al. (2014)*. Two-day-old light-grown seedlings were treated with His$_6$-RALF1 for 2 days according to *Haruta et al. (2014)* at concentrations indicated in the figures. Root lengths were measured at the beginning and end of treatments to obtain growth during treatment. Epidermal cell analysis was carried out as described (*Le et al., 2006*; *Sorek et al., 2011*). ROS in the primary roots and root hairs were detected by H$_2$DCF–DA (2′, 7′-dichlorodihydro-fluorescein diacetate; Sigma/Aldrich, St. Louis, MO) and ROS fluorescence intensity within a fixed region of interest (ROI) was quantified using Image J.

## Ovule analysis

FER-GFP localization in ovules was acquired as described in *Duan et al. (2014)*. The synergid cells from one ovule to another are not identical in shape, size, or relative orientation with the rest of the ovule parts. For comparative quantitative analysis of data between wild type and mutant ovules, signals from the entire recognizable filiform apparatus and synergid cells were quantified and relative signal distribution between the filiform apparatus and the synergid cell cytoplasm was compared between wild type and mutant ovules.

## Protein–protein interaction assays

For protein pull-downs, bait proteins (MBP-LLG1, MBP-LRE1, MBP-ROP2, MBP-exJM, His$_6$-LLG1, MBP-RALF1) were produced in *E. coli* and bound to amylose or talon resins as previously described (*Duan et al., 2010*). Plant proteins (FER-HA, FER-GFP, FERΔexJM-GFP, HA-LLG1, RbohD(N)-HA) were expressed in protoplasts (≤10 μg DNA per transfection) and extracted at ~12 hr after transfection in pull-down buffer (binding buffer: 40 mM Tris–HCl, pH 7.5, 100 mM NaCl, 1 mM

Na₂-EDTA; plus 5% glycerol, 5 mM $MgCl_2$, 1 mM PMSF, protease inhibitor mixture [Calbiochem, San Diego, CA] at 1:100 dilution, and 0.4% Triton X-100 to facilitate solubilization). Protoplast protein extracts or *E. coli*-produced target proteins were applied to bait protein-bound resins and incubated at 4°C for 2 hr with gentle mixing. The resin was washed three times in binding buffer. Proteins remained bound to the resin were eluted by mixing with SDS/PAGE loading buffer, boiled for 5 min, and applied to SDS/PAGE (7.5% for FER; 12.5–17.5% for LLG1 and LRE) for immunoblot analysis. Protein blots were stained by Ponçeau S Sigma-Aldrich for sample loading comparison, followed by immunostaining. Primary (anti-HA and anti-GFP) and secondary antibodies for chemiluminescence detection were from Santa Cruz. Signals were acquired by the PXi imaging system (Syngene, Cambridge, UK).

MBP-ROP2 pull-down of protoplasts-expressed HA-LLG1 and HA-LRE followed the previously described procedure (*Duan et al., 2010*). Pull-down of RbohD(N)-HA was carried out similarly with MBP-ROP2 resin pretreated by 10 mM GTP or GDP for 2 hr and pull-down carried out with 10 mM GTP or GDP in the buffer.

For co-immunoprecipitation, *35S::HA-LLG1* and *35S::FER-GFP* were co-expressed in transfected protoplasts. Mock samples were transfected with *35S::FER-GFP* and empty SK vector DNA. Proteins were extracted as described above. Anti-HA antibody was used at 1:100 dilution for each immunoprecipitation in 1 ml reactions, incubated at 4°C for 3 hr, followed by the addition of 50 µl of protein G resin suspension (Santa Cruz Technology, Dellas, TX.). After binding for 1 hr, the resin was washed five times in binding buffer. Proteins remained bound to the resin were eluted in SDS/PAGE loading buffer, boiled for 5 min, and applied to SDS/PAGE for immunoblot analysis as described above.

For BiFC, the split-VENUS system (*Kodama and Hu, 2012*) was used for Arabidopsis protoplast transfection (*Duan et al., 2010*) and agroinfiltration of tobacco leaf epidermis (*Batoko et al., 2000*).

For yeast two-hybrid assays, products and procedures from Stratagene were used for vectors, yeast growth, selection, and β-galactosidase activity.

## Microscopy

Seedling, inflorescence, and trichome images were acquired on an Olympus SZ61 microscope. Epifluorescence and DIC microscopy were carried out on a NIKON Eclipse E800 microscope equipped with a SPOT camera (Molecular Diagnostic). Filters from Chroma were used: green fluorescence, Ex460-500/DM505/BA510-560; red fluorescence, Ex546(10)/ DM565LP/EM590LP; yellow fluorescence, Ex490-510/DM515/BA520-550. Confocal imaging was carried out on a Zeiss Meta510 or a Nikon A1. Comparative studies were based on identical imaging conditions and followed procedures described in *Duan et al. (2010)*. Use of ER-Tracker Red (BODIPY TR Glibenclamide; Life-Technologies) to stain ER followed *Cui et al. (2012)*, *Yang et al. (2013)*, and the manufacturer's protocol.

## Data analysis and presentation

All data presented are representative of at least three independent experiments with comparable results. Quantitative data are presented as averages ± SEM of replicated experiments or samples, or as averages ± SD from a representative experiment as indicated in the figure legends. Sampling sizes are indicated in the figures or figure legends. Student's *t*-tests were used for p value calculations. $p < 0.05$ is considered significant (indicated by * in figures); most experimental and control data were significantly different with p values ranging from $10^{-2}$ to $10^{-5}$ (indicated by ** in figures).

## Acknowledgements

We thank Ravi Palanivelu (University of Arizona, Tucson) for sharing *lre-5* and *llg1-1* mutants and for communications; Jaideep Mathur (Guelph University, Toronto) for *35S-RFP-ER* construct. MCL was supported by a fellowship from the National Research Council of Taiwan. This work was supported by grants from the National Science Foundation (IOS-1127002 and IOS-1146941).

Author contributions. CL contributed to a majority of the experiments; others obtained and analyzed experimental data to complete the study. HMW and AYC led the overall research design; HMW made all molecular constructs. HMW, AYC, and CL wrote the manuscript; FLY, QD, DK, MCL, and BWW contributed to the process. All authors read and approved the manuscript.

# Additional information

## Funding

| Funder | Grant reference | Author |
|---|---|---|
| National Science Foundation (NSF) | IOS-1127002 | Alice Y Cheung, Hen-Ming Wu |
| National Science Foundation (NSF) | IOS-1146941 | Hen-Ming Wu |

The funder had no role in study design, data collection and interpretation, or the decision to submit the work for publication.

## Author contributions

CL, F-LY, AYC, QD, DK, M-CL, H-MW, Conception and design, Acquisition of data, Analysis and interpretation of data, Drafting or revising the article; JM, Conception and design, Acquisition of data, Analysis and interpretation of data; EJL, LG, MJ, AK, HC, Acquisition of data, Analysis and interpretation of data; BWW, Acquisition of data, Analysis and interpretation of data, Drafting or revising the article

## Author ORCIDs

Alice Y Cheung, http://orcid.org/0000-0002-7973-022X

# Additional files

## Supplementary file

• Supplementary file 1. A list of gene constructs.

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
