## [Decision Letter]

Thank you for sending your work entitled “Glycosylphosphatidylinositol-anchored proteins as co-receptors for FERONIA receptor kinase signaling in *Arabidopsis*” for consideration at *eLife*. Your article has been favorably evaluated by Detlef Weigel (Senior editor) and three reviewers, one of whom is a member of our Board of Reviewing Editors.

The following individuals responsible for the peer review of your submission have agreed to reveal their identity: Sheila McCormick (Reviewing editor and reviewer) and Venkatesan Sundaresan (reviewer). A further reviewer remains anonymous.

The Reviewing editor and the other reviewers discussed their comments before we reached this decision, and the Reviewing editor has assembled the following comments to help you prepare a revised submission.

The FER kinase is involved in several signaling pathways that are important for plant development, so the problem investigated is of high interest. The findings provide new insights into the mechanism of signaling by FER, and into the molecular functions of the LRE-related GPI-anchored proteins, which are poorly understood in plants. Overall the study appears be well-executed and the data are solid. Evidence for LLG1 functioning as a chaperone for FER delivery to the cell membrane is robust, but evidence for it functioning as a co-receptor for signal perception is weaker. Additionally, we think the manuscript can be improved (e.g., sentence structure, better explanation of the logic behind each experiment), as in many places it was difficult to follow or information was missing.

1) Clarifications and additional experiments needed to support the model. The proposed model is that LRE/LLG1 forms a tetrameric signaling complex with FER and inactive RAC/ROP proteins. The FER- LRE/LLG interactions are supported by yeast 2-hybrid as well as BIFC studies and are convincing that LLG1 is a chaperone for localization of FER to the PM. However, several experiments are needed to strengthen the conclusion about LLG1's role as a co-receptor: (A) FER-mediated molecular responses to RALF1, such as gene expression, should be analyzed in *fer* and *llg1* mutants (e.g. the TCH and BR6ox genes reported in Shih et al., and [23]). (B) Does RALF1 alter FER-LLG1 interaction, e.g. in the co-IP assays shown in Figure 8? (C) Is RALF1 binding to FER affected in the *llg1* mutant (as assayed in [23])? In other words, if LLG1 + FER bind RALF, but LRE + FER do not (based on the phenotypes of *llg1* and *lre*), that would be strong support of the model that FER selectivity is due to association with LRE-like proteins. As we understand it, the ROP2 pulldown was performed with LLG1 and LRE expressed in protoplasts (not *E. coli*), which should have the endogenous FER. Therefore, the likely explanation is that FER mediated the pulldown. But then this should be tested using the *fer* mutant and FER co-overexpression, and also in the presence and absence of RALF1 peptide. That is, *fer* mutant protoplasts transformed with (1) LLG1-HA, (2) LLG1-HA + 35S-FER, and (3) LLG1-HA+35S-FER + RALF1 treatment. These three samples can potentially answer several key questions raised by their model. Furthermore, whether LLG1 function is required for the FER-ROP2 interaction can be tested by doing the ROP2-pulldown-FER assay in *llg1* mutant protoplasts. Their model (Figure 11) proposes the “signal” (somehow) causes RACROP disassociation from the receptor complex and could be tested by including RALF1 in their ROP pulldown assays. These experiments can potentially add significant insights and clarify some important uncertainties of their model.

2) Clarifications re phenotypic characterization: Phenotypes were compared in detail between *llg1* and *fer*, but double mutant and epistatic analyses were not performed. Although an additive effect can be caused by partial loss of function (with redundant genes present), absence of an additive effect between *llg1* and fer would provide key support for their function in the same pathway. If there is a reason why double mutants were not analyzed, it should be explained. [23] and [47] showed longer roots in *fer* than in wild type, opposite to the results shown in Figure 4. This requires clarification! While the RALF1 insensitive root of *llg1* supports the requirement of LLG1 for RALF1 signaling, the retarded root growth of untreated *llg1* mutant, similar to RALF1-treated wild type, is consistent with constitutive RALF1/FER signaling.

3) Microscopic data need clarification. For example, the “well-defined cell membrane localization” and “mislocalized… in *llg1* mutant” in Figure 6 is not obvious, because the signal intensity in WT is lower than that in *llg1*, and the cell membrane still shows a stronger signal than the ER in *llg1*. A similar amplified view should be provided for WT (there is empty space in the figure layout). In Figure 6, is the effect specific to FER or would any cell surface receptor be affected by this root hair tip defect? What is the difference between WT and *llg1* in Figure 6—figure supplement 1? Figure 6—figure supplement 1 lack wild type and non-transgenic controls for the “auto-green fluorescent patches”. Figure 6—figure supplement 2: why is the signal intensity different between WT and *lre* and *lre llg1*? And why in the *lre* mutant gametophytes, is FER-GFP localization cytoplasmic? Wouldn't the FER-GFP signal be expected in the ER of the synergids, similar to the ER localization of FER-GFP in the hypocotyl cells of *llg*1 mutants? Furthermore, the ratios of FER in the FA vs. outside the FA in the synergids are estimated based on integrating the GFP signal over a ROI. It is not clear how the ROIs were selected, because the outline of the synergids is not clear. The ROIs for the mutants seem larger, especially in the case of the *lre llg1* double mutant. This might result in an artificially high value for the signal outside the FA in the mutant gametophytes. A justification for the ROIs selected should be provided. Figure 7: would be more convincing and clear if random views of multiple protoplasts were shown.

4) The *lre* mutants used in the study should be described in the Methods. In the case of *lre*, which is a female gametophyte mutant, the plants might be assumed to be heterozygous for the mutation. Based on the frequencies in Figure 6, it would appear that the *lre* plants are in fact homozygous, but it is not obvious. In general, the Methods section should clarify the homozygosity of all the mutants used.

---

## [Author Response]

*1) Clarifications and additional experiments needed to support the model. The proposed model is that LRE/LLG1 forms a tetrameric signaling complex with FER and inactive RAC/ROP proteins. The FER- LRE/LLG interactions are supported by yeast 2-hybrid as well as BIFC studies and are convincing that LLG1 is a chaperone for localization of FER to the PM. However, several experiments are needed to strengthen the conclusion about LLG1's role as a co-receptor: (A) FER-mediated molecular responses to RALF1, such as gene expression, should be analyzed in* fer *and* llg1 *mutants (e.g. the TCH and BR6ox genes reported in*
[47]
*and*
[23]*)*.

We tested several RALF1-regulated genes and confirm that *llg1-2* and *fer-4* were similar in their reduced sensitivity relative to wild type (data added as Figure 4, described in the subsection headed “*llg1* and *fer* mutants have indistinguishable hormone- and RAC/ROP-regulated phenotypes”). [23] showed only that BR6OX2 expression in *fer-4* was less sensitive to RALF1 suppression than WT. We extended the analysis to include four other RALF1 regulated genes, and showed that while RALF1 suppressed BR6OX2 and GA3OX3 and stimulated CML38 and ERF6, *fer-4* and *llg1-2* were both less sensitive to RALF1-induced fluctuations. We did not examine TCH, as we are aware of another lab's focus on this aspect in llg1, so do not wish to knowingly intrude on that.

*(B) Does RALF1 alter FER-LLG1 interaction, e.g. in the co-IP assays shown in*
Figure 8*? (C) Is RALF1 binding to FER affected in the* llg1 *mutant (as assayed in*
[23]*)? In other words, if LLG1 + FER bind RALF, but LRE + FER do not (based on the phenotypes of* llg1 *and* lre*), that would be strong support of the model that FER selectivity is due to association with LRE-like proteins. As we understand it, the ROP2 pulldown was performed with LLG1 and LRE expressed in protoplasts (not E. coli), which should have the endogenous FER. Therefore, the likely explanation is that FER mediated the pulldown. But then this should be tested using the* fer *mutant and FER co-overexpression, and also in the presence and absence of RALF1 peptide. That is,* fer *mutant protoplasts transformed with (1) LLG1-HA, (2) LLG1-HA + 35S-FER, and (3) LLG1-HA+35S-FER + RALF1 treatment. These three samples can potentially answer several key questions raised by their model. Furthermore, whether LLG1 function is required for the FER-ROP2 interaction can be tested by doing the ROP2-pulldown-FER assay in* llg1 *mutant protoplasts. Their model (*Figure 11*) proposes the “signal” (somehow) causes RACROP disassociation from the receptor complex and could be tested by including RALF1 in their ROP pulldown assays. These experiments can potentially add significant insights and clarify some important uncertainties of their model*.

The list represents a comprehensive set of biochemical studies to unravel the FER-LLG1/LRE and RALF1 relationship which are in fact ongoing projects in our lab. Although we explained above that RALF1 is not the only concern in this study, we add to the revised manuscript a result showing that RALF1 pulled-down co-expressed FER and LLG1 (added as Figure 8; please see the subsection headed “FER interacts with LLG1 on the cell membrane and in the ER”), supporting interaction between RALF1 and the FER-LLG1 complex. Together with the expanded results showing comparable reduction of sensitivity to RALF1-regulated gene expression (response above) and lack of additive effects of combining *fer4* and *llg1* (response below), we believe that the current version contains data that considerably strengthen the FER-LLG1 co-receptor concept.

As for the list of experiments suggested by the reviewers, investigations along similar lines were already ongoing (by F-L. Yeh, and J.M.) as we wrapped up the Li et al. study. Every question raised is important but answering each already involves relatively complex experimentation. We do not wish to treat any of these aspects without the thoroughness we like to maintain in our work or without being able to take into account the multiple facets of FER functions. Given the already very extensive study presented in Li et al., we feel that we cannot do justice to the depth that these studies requires or to the junior investigators who carry out these works by hastily putting results from ongoing work into this paper.

*2) Clarifications re phenotypic characterization: Phenotypes were compared in detail between* llg1 *and* fer*, but double mutant and epistatic analyses were not performed. Although an additive effect can be caused by partial loss of function (with redundant genes present), absence of an additive effect between* llg1 *and* fer *would provide key support for their function in the same pathway. If there is a reason why double mutants were not analyzed, it should be explained*.

Yes, double *fer-4 llg1-2* seedling phenotype mimic those in individual parents (added as Figure 2—figure supplement 2; please also see the subsection “*llg1* and *fer* mutants have indistinguishable growth and developmental phenotypes”), providing additional and key support for FER and LLG1 functioning in the same pathway.

*Haruta et al. and Shih et al. showed longer roots in* fer *than in wild type, opposite to the results shown in*
Figure 4*. This requires clarification! While the RALF1 insensitive root of* llg1 *supports the requirement of LLG1 for RALF1 signaling, the retarded root growth of untreated* llg1 *mutant, similar to RALF1-treated wild type, is consistent with constitutive RALF1/FER signaling*.

We explain the growth phenotype in more detail in the second and third paragraphs of this response and in the text (subsection entitled “*llg1* and *fer* mutants have indistinguishable hormone- and RAC/ROP-regulated phenotypes”). We respond first on the comments on Haruta et al. and Shih et al.

Regarding Haruta et al. and Shih et al. results. We do not agree with the assessment that “Haruta et al. and Shih et al., 2014 showed longer roots in *fer* than in wild type”. If one examines Haruta et al. Figure 2, mock-treated *fer-4* roots were shorter than wild type ones (compare first and second data sets in the plot). The seedling image in Figure 2, one of the *fer-4* seedlings was longer, the other was shorter or at most comparable to wild type, suggesting that the illustrated images were likely among the longer population of *fer-4,* which are routinely the better survivors. As for the Shih et al. study, the main text and SI together showed two partial *fer-4* root images (Figure 2, and S2E). The more severely growth-retarded seedlings also have strong cell wall defects, including rips between cells (D. Kita, Ph.D. dissertation, and manuscript under preparation) and, in our opinion, would not have been good candidates for assessment of mechanical sensing. Judging from the notable root hairs still observable in *fer-4* in Shih et al. Figure 2, plants used in the mechanical sensing studies would indeed have been the larger-sized seedlings, possibly approximating wild type. However, the overall plant sizes in Shih et al. could not be evaluated from the results shown in the paper.

Detail description of *fer-4* growth and to clarify the RALF regulated growth phenotype*:* The vegetative phenotypes of *fer-4* and *llg1* and the reproductive phenotypes (i.e. female gametophyte sterile) of *fer-4* and *lre-5* are not 100% penetrant. Seedlings vary in sizes (see added Figure 2—figure supplement 2) and only a majority of *fer-4* and *lre-5* female gametophytes are sterile but a percentage remains fertile, producing homozygous mutant progeny (17; 16; 52). All *fer-4* phenotypes are complemented by FER-GFP (17; 16). The most severe *fer-4* seedlings do not survive, but the same range of phenotypes bled true in progeny from less severely defective mutant plants. We tested another allele, *fer-2* (15), the non-penetrant phenotype is the same. *llg1-1* and *llg1-2* seedlings are also heterogeneous in sizes.

Larger and smaller *fer* and *llg1* seedlings persisted in RALF treatment samples (added as Figure 4—figure supplement 2), consistent with reduced sensitivity to RALF in the overall mutant seedling populations. The size (thus growth) heterogeneity led to our measuring “actual growth” during the two days of treatment and using the smaller seedlings (Figure 4 as in original manuscript) because they were more homogeneous. We believe the actual growth measurements adds strength to just comparing root lengths of wild type and mutant seedlings after treatments (as in Haruta et al., and as we also show in Figure 4, Figure 4—figure supplement 2, right).

*3) Microscopic data need clarification. For example, the “well-defined cell membrane localization” and “mislocalized… in* llg1 *mutant” in*
Figure 6
*is not obvious, because the signal intensity in WT is lower than that in llg1*.

The signal intensity is actually not directly comparable in these images, because they were captured by “autoexposure” and the exposure time is determined by the highest signal intensity in the overall sample. In fact, FER-GFP signals were consistently not notably different between wild type and *llg1-2*; this is also evidenced here by the comparable signal to background contrast.

*And the cell membrane still shows a stronger signal than the ER in* llg1*.*

The point we wanted to convey is that there was considerably higher retention of FER-GFP in *llg1* tissues but not that it was entirely removed from the cell periphery (where the cell membrane and cortical ER are often not easily resolved). We change the sentence that refers to this result to: “intracellular FER-GFP signal was considerably more pronounced in *llg1* hypocotyl (Figure 6) and roots (Figure 6; Figure 6—figure supplement 1) than in wild type tissues”.

*A similar amplified view should be provided for WT (there is empty space in the figure layout)*.

Done (added to Figure 6).

*In*
Figure 6*, is the effect specific to FER or would any cell surface receptor be affected by this root hair tip defect?*

We opt to leave this question open for this study, and added a sentence in the Discussion: “To what extent the LRE family protein controls the delivery of receptor kinases as a class or more specifically those closely related to FER remains to be determined”.

On a practical level, we cannot exhaustively examine all, or even a large number of cell surface receptors, so cannot generalize “specificity” for FER even if we report on a very small subset of them. On the other hand, *fer* mutants have multiple signaling phenotypes (e.g. auxin, ethylene, ABA, brassinosteroid, pathogen, mechanical sensing). Beyond the phenotype reported here, unpublished observations from our lab indicate that *llg1* has similar brassinosteroid and ethylene-related phenotypes as *fer-4*. So, it is quite plausible that additional cell surface signaling molecules can be impacted by loss of LLG1. For these reasons, we would like to leave this question for later reporting.

*What is the difference between WT and* llg1 *in*
Figure 6—figure supplement 1*?*

We replace the original wide-field pictures with maximum projections from confocal stacks from similar plants. They are meant to show considerably higher intracellular FER-GFP signal in the *llg1* root segment relative to in wild type roots as indicated in the text and figure legend.

Figure 6—figure supplement 1
*lack wild type and non-transgenic controls for the “auto-green fluorescent patches”*.

We added a more relevant control, a comparable optical section of a FER-GFP in wild type (Figure 6—figure supplement 1, right) where little extraneous fluorescence existed. Wild type and non-transgenic controls under similar image acquisition conditions would be just black.

The fact is both *fer-4* and *llg1* have severe cell surface defects, one of these being having random patches of darkened tissues that are also highly fluorescent (shown below), perhaps reflecting cell death.

Author response image 1.**DOI:**
http://dx.doi.org/10.7554/eLife.06587.032

Figure 6—figure supplement 2: *why is the signal intensity different between WT and* lre *and* lre llg1*?*

These images were acquired by autoexposure. In wild type ovules, FER-GFP is focused in the filiform apparatus (FA) (very densely packed with cell membrane); FER-GFP is also expressed in other ovular cells but under the exposure conditions the exposure time was short and the contrast between the peak signal at the FA and the rest was very high. In *lre-5* and *lre-5 llg1-2* ovules, FER-GFP is not focused in the cell membrane but distributed throughout the synergid cell (and so diluting the per pixel fluorescent signal), requiring longer exposure time, which then also revealed signal from other ovular cells.

*And why in the* lre *mutant gametophytes, is FER-GFP localization cytoplasmic? Wouldn't the FER-GFP signal be expected in the ER of the synergids, similar to the ER localization of FER-GFP in the hypocotyl cells of* llg1 *mutants?*

The synergid cells are highly secretory and the cytoplasm is packed with endomembrane but usually cannot be resolved into clear reticulate patterns under widefield imaging, though sometimes they did appear in patches (as seen in Figure 6, right image; it is now also shown magnified in Figure 6—figure supplement 2; please see subsection headed “Loss of LLG1 and LRE functions suppresses FER localization to the cell membrane”). The signal peptide targets FER-GFP into the ER as it is translated and should remain associated with the endomembrane system. Even with that expectation, we stated in the manuscript only that “a significantly higher percentage of (*lre*) mutant ovules relative to the wild type showed female gametophyte FER-GFP either totally retained in the synergid cell cytoplasm…”.

*Furthermore, the ratios of FER in the FA vs. outside the FA in the synergids are estimated based on integrating the GFP signal over a ROI. It is not clear how the ROIs were selected, because the outline of the synergids is not clear. The ROIs for the mutants seem larger, especially in the case of the* lre llg1 *double mutant. This might result in an artificially high value for the signal outside the FA in the mutant gametophytes. A justification for the ROIs selected should be provided*.

The synergid cell (pink) and the filiform apparatus (white) are more clearly drawn in the revised version (Figure 6—figure supplement 2). The synergid cells from one ovule to another are not identical (not even similar) in shape, size and relative orientation with the rest of the ovule parts. The ROIs are therefore the entire areas recognizable as the FA and the synergid cells. These size and shape differences are another reason that relative average intensity/pixel in the FA: that in the synergid cell cytoplasmic signals were compared instead of total intensity. This description has been added to the Methods section (in the subsection headed “Ovule analysis”).

Figure 7*: would be more convincing and clear if random views of multiple protoplasts were shown.*

To discern cellular location, we had to use 100x lens, which almost never had more than a single transformed cell in one frame (due to the relatively low % of cells showing FER-GFP signal at the amount of input transgenes used). We have some lower magnification images showing a few cells, which are suggestive of notable differences but not adequately clear as to the precise cellular location. We added these as Figure 7—figure supplement 1 (referred to in the subsection headed “Loss of LLG1 induces retention of FER in the ER”).

*4) The* lre *mutants used in the study should be described in the Methods. In the case of* lre*, which is a female gametophyte mutant, the plants might be assumed to be heterozygous for the mutation. Based on the frequencies in*
Figure 6*, it would appear that the* lre *plants are in fact homozygous, but it is not obvious. In general, the Methods section should clarify the homozygosity of all the mutants used*.

Done, we indicate in the first paragraph of the Methods section that *fer-4* and *lre-5* were homozygous. Although both *fer* and *lre* were described as female gametophytic mutants, both actually produce seeds and homozygous progeny exists (see [17]; [16]; [52]).